# Short senolytic or senostatic interventions rescue progression of radiation-induced frailty and premature ageing in mice

Edward Fielder†, Tengfei Wan†, Ghazaleh Alimohammadiha, Abbas Ishaq§, Evon Low, B Melanie Weigand#, George Kelly, Craig Parker, Brigid Griffin, Diana Jurk#, Viktor I Korolchuk, Thomas von Zglinicki*‡, Satomi Miwa‡

Newcastle University Biosciences Institute, Newcastle University, Newcastle upon Tyne, Newcastle, United Kingdom

**\*For correspondence:**
t.vonzglinicki@ncl.ac.uk

†These authors contributed equally to this work
‡These authors also contributed equally to this work

**Present address:** §Alcyomics Ltd, The Biosphere, Newcastle Helix, Drayman's Way, Newcastle, United Kingdom; #Robert and Arlene Kogod Center on Aging, Department of Physiology and Biomedical Engineering, Mayo Clinic, Rochester, United States

**Abstract** Cancer survivors suffer from progressive frailty, multimorbidity, and premature morbidity. We hypothesise that therapy-induced senescence and senescence progression via bystander effects are significant causes of this premature ageing phenotype. Accordingly, the study addresses the question whether a short anti-senescence intervention is able to block progression of radiation-induced frailty and disability in a pre-clinical setting. Male mice were sublethally irradiated at 5 months of age and treated (or not) with either a senolytic drug (Navitoclax or dasatinib + quercetin) for 10 days or with the senostatic metformin for 10 weeks. Follow-up was for 1 year. Treatments commencing within a month after irradiation effectively reduced frailty progression (p<0.05) and improved muscle (p<0.01) and liver (p<0.05) function as well as short-term memory (p<0.05) until advanced age with no need for repeated interventions. Senolytic interventions that started late, after radiation-induced premature frailty was manifest, still had beneficial effects on frailty (p<0.05) and short-term memory (p<0.05). Metformin was similarly effective as senolytics. At therapeutically achievable concentrations, metformin acted as a senostatic neither via inhibition of mitochondrial complex I, nor via improvement of mitophagy or mitochondrial function, but by reducing non-mitochondrial reactive oxygen species production via NADPH oxidase 4 inhibition in senescent cells. Our study suggests that the progression of adverse long-term health and quality-of-life effects of radiation exposure, as experienced by cancer survivors, might be rescued by short-term adjuvant anti-senescence interventions.

## Editor's evaluation

This is an exciting study with translational implications that patients exposed to radiation as a form of therapy for cancer may avoid complications down the road by utilising medication to eliminate the senescent cells generated by the treatment and improve quality of life.

## Introduction

Cancer has become an increasingly survivable disease, with cancer-specific mortality in developed countries having dropped sharply in the last few decades. Many cancer types have now high cure rates (*Bray et al., 2018*), and in some fields the emphasis has started to shift towards efforts to improve the quality of survivorship after successful cancer treatment (*Damlaj et al., 2019*).This is necessary as long-term survivors of childhood and adult cancers undergo a wide-range of negative health and

**eLife digest** Cancer treatments save lives, but they can also be associated with long-term side effects which greatly reduce quality of life; former patients often face fatigue, memory loss, frailty, higher likelihood of developing other cancers, and overall accelerated aging.

Senescence is a change in a cell's state that follows damage and is associated with aging. When a cell becomes senescent it stops dividing, can promote inflammation and may damage other cells. Research has shown that cancer treatment increases the numbers of cells entering senescence, potentially explaining the associated long-term side effects. A new class of drugs known as senolytics can kill senescent cells, but whether they could help to counteract the damaging effects of cancer treatments remain unclear.

To explore this question, Fielder et al. focused on mice having received radiation therapy, which also exhibit the long-term health defects observed in human patients. In these animals, a single, short senolytic treatment after irradiation nearly erased premature aging; frailty did not increase faster than normal, new cancers were less prevalent, and the rodents retained good memory and muscle function for at least one year after irradiation. Even mice treated later in life, after frailty was already established, showed some improvement. In addition, multiple tissues, including the brain and the liver, hosted fewer senescent cells in the animals treated with senolytics, even up to old age. Research should now explore whether these remarkable effects could also be true for humans.

quality-of-life changes that lead to increased frailty, multimorbidity, and mortality compared to the general population (*Bluethmann et al., 2016*; *Cupit-Link et al., 2017*; *Robison and Hudson, 2014*). These changes are indicative of accelerated or premature ageing in long-term cancer survivors, for which there is currently no validated therapy.

Premature ageing in cancer survivors appears to be largely caused by DNA-damaging cancer therapies. Numerous biological processes have been proposed as drivers of this (*Cupit-Link et al., 2017*; *Ness et al., 2018*), with therapy-induced cell senescence prominent amongst them (*Short et al., 2019*).

Cell senescence is a complex cellular stress response programme that can be induced by DNA damage (e.g. radio- or chemotherapy) and involves persistent cell cycle arrest, aberrant regulation of metabolism (specifically energy metabolism), epigenetic programming, and secretory processes (*Gorgoulis et al., 2019*; *von Zglinicki, 2021*). Therapy-induced senescence may constitute a cytostatic clinical response contributing to stable disease (*te Poele et al., 2002*); however, there is increasing evidence that therapy-induced senescent cells can promote both primary relapse and secondary cancers, often leading to less successful treatment outcomes of consequent disease (*Jena et al., 2020*; *Saleh et al., 2019*). Senescent cells release both pro-inflammatory and pro-oxidant signalling molecules (the senescence-associated secretory phenotype [SASP]) which can damage and induce senescence in bystander cells (*Nelson et al., 2012*), and thus spread the phenotype from the point of origin throughout tissues and organisms (*da Silva et al., 2019*; *Xu et al., 2018*). As such, adjuvant tumour therapy not only induces transiently (and locally) high concentrations of senescent cells but may also result in faster accumulation of these cells both locally and systemically over the whole life course (*Short et al., 2019*).

Accumulation of senescent cells is causal for a wide range of ageing-associated diseases and disabilities as evidenced by the far-reaching successes of interventions that reduce the systemic load of senescence [for review see *Short et al., 2019*]. In fact, acute ablation of senescent cells by continuous pharmacogenetic or pharmacologic intervention has been able to reduce chemotherapy-induced multimorbidity (*Demaria et al., 2017*) and liver toxicity (*Baar et al., 2017*) as well as radiation-induced haematoxicity (*Chang et al., 2016*) and sarcopaenia (*Zhu et al., 2015*) in mice. However, if induction of secondary senescence by bystander effects is a major driver of post-therapeutic senescence, continuous anti-senescence interventions might not be necessary. Rather, we hypothesised that specific ablation of senescent cells (senolytic intervention) or specific inhibition of the SASP (senostatic intervention) shortly after adjuvant cancer therapy might be sufficient to rescue enhanced mortality, multimorbidity, and frailty in cancer survivors over their life course (*Short et al., 2019*).

Senolytics are potent drugs with frequently serious side effects (*Demaria, 2017*) that would raise significant safety concerns in a preventive setting, even in a high-risk group like tumour survivors. In contrast, senostatics (sometimes also termed senomorphics) are chemicals that do not kill (senescent) cells but block SASP signals, thus inhibiting the spread of senescence via bystander effects. In fact, senostatic interventions, including the dietary restriction mimetics rapamycin and metformin, or dietary restriction itself, caused lasting reductions of senescent cell burden in different tissues of mice with improved lifespan and healthspan (*Blagosklonny, 2017*; *Fontana et al., 2018*; *López-Otín et al., 2016*; *Selvarani et al., 2021*). However, their efficiency to rescue premature ageing has not been pre-clinically tested in comparison to senolytic intervention.

The senostatic metformin has an extraordinarily good safety profile, which has been testified in its long clinical history as well as in a myriad of clinical trials. However, the drug acts through multiple pathways, and it is not at all clear how it reduces the SASP in a therapeutically achievable setting. It is often assumed that it blocks complex I of the electron transport chain, thus reducing production of reactive oxygen species (ROS) in mitochondria, which in turn would reduce NF-κB activation and thus the SASP (*Moiseeva et al., 2013*). However, metformin efficiently blocks complex I only in millimolar concentrations, while tissue concentrations that can be achieved in mice or man are one to two orders of magnitude lower (*Wilcock and Bailey, 1994*). A mechanistic examination of the senostatic effect of metformin in vivo is therefore urgently warranted.

To address these questions in a pre-clinical setting, we used a simple mouse model of premature ageing induced by fractionated whole-body irradiation (*Fielder et al., 2019*). For proof of principle, we used first generation senolytics with well-documented efficacy (Navitoclax and dasatinib + quercetin) in a wide range of age-associated deficiencies and tissues. We focussed on frailty as a primary outcome because premature frailty is a well-documented, clinically important problem of long-term cancer survivors (*Ness et al., 2013*) as well as a strong predictor of multimorbidity and mortality in humans (*Kojima et al., 2018*) and mice (*Whitehead et al., 2014*). Sarcopaenia, metabolic disease, and, especially, cognitive decline (*Országhová et al., 2021*) are major complications in long-term tumour survivors, therefore we included phenotype assays in these domains combined with tissue-specific ex vivo analyses primarily in hippocampus, muscle, and liver. We show here that (i) irradiation-induced lifelong premature ageing can be rescued by a one-off post-irradiation senolytic intervention, (ii) senolytics are still partially efficient in reducing progression after establishment of a premature ageing phenotype, (iii) a relatively short metformin intervention is similarly effective in rescuing premature ageing, and (iv) metformin at therapeutic concentrations acts as a senostatic neither via inhibition of complex I, nor via improvement of mitophagy or mitochondrial function, but by reducing non-mitochondrial ROS production in senescent cells.

## Results

### Short-term post-irradiation senolytic interventions rescue premature ageing

Male C57Bl/6 J mice received fractionated sublethal whole-body irradiation (IR, 3 × 3 Gy) at an age of 5–6 months and were treated with a short course (10 days) of senolytics by oral gavage at 1 month after irradiation, i.e., when acute radiation sickness had abated (*Figure 1A*). Doses were 5 mg/kg/day dasatinib and 50 mg/kg/day quercetin (D+Q) as typically applied in senolytic mouse studies (*Palmer et al., 2019*; *Schafer et al., 2017*; *Zhu et al., 2015*). As a senolytic, Navitoclax has been used at widely different doses in mice, ranging from 1.5 mg/kg/day (*Tarantini et al., 2021*; *Yabluchanskiy et al., 2020*) to 50 mg/kg/day (*Bussian et al., 2018*; *Chang et al., 2016*). Because Navitoclax causes thrombocytopenia at higher concentrations (*Wilson et al., 2010*), we decided to use Navitoclax in the lower concentration range, e.g., at 5 mg/kg/day. Irradiated mice experienced premature ageing as documented by a doubling of the rate of frailty progression, decreased neuromuscular co-ordination, decreased short-term memory, and increased general and cancer-associated mortality (*Fielder et al., 2019* and *Figure 1*). When mice were treated with either senolytic drug at 1 month after IR indicators of premature ageing were rescued over almost 1 year of follow-up (*Figure 1B – G*). Although frailty was not reversed, rates of frailty progression decreased after senolytic treatment to values comparable to non-irradiated mice (*Figure 1B*). The frailty index is composed of 30 different assessments. Early intervention with each of the senolytics improved six of them, namely, mouse grimace

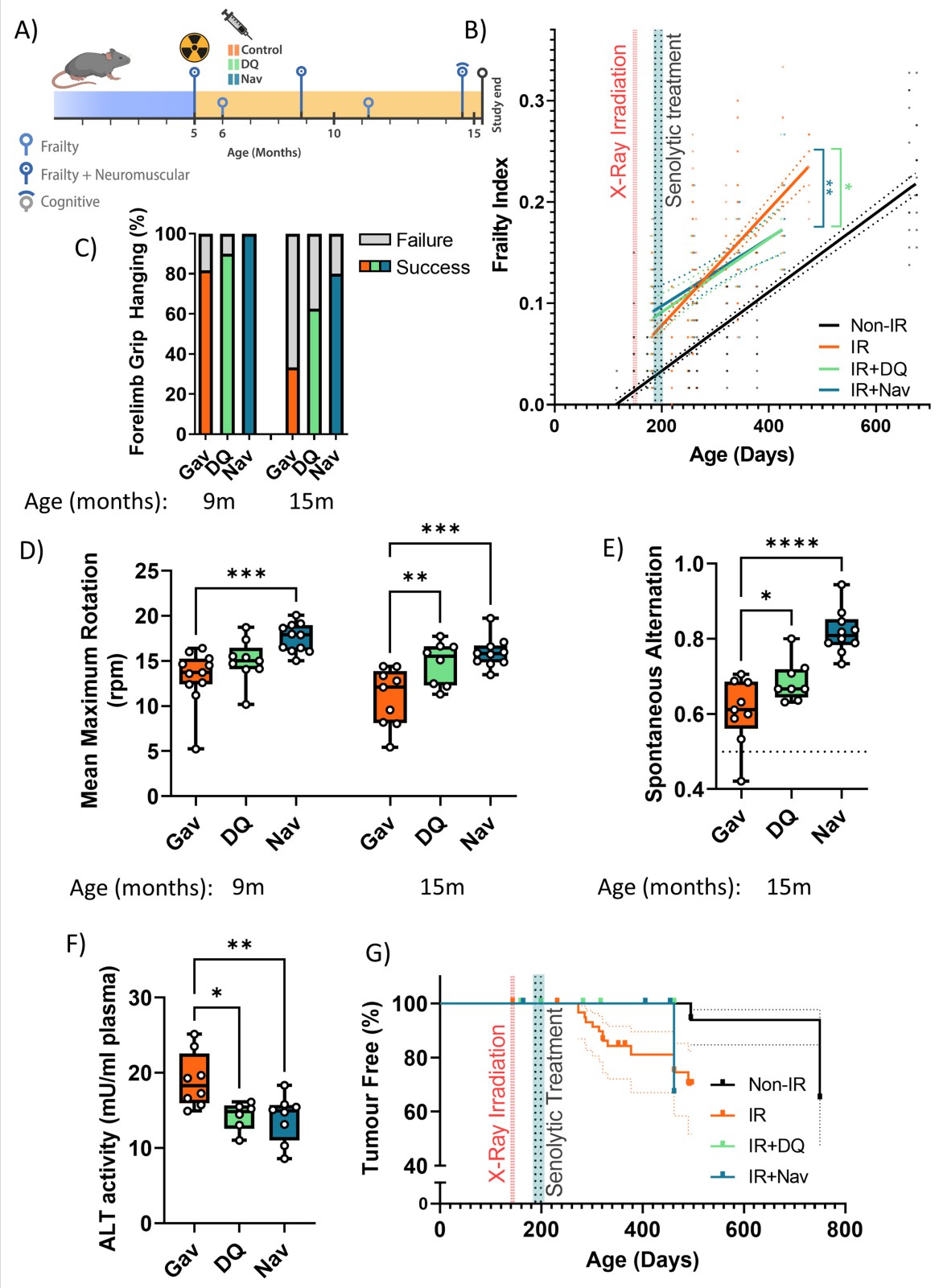

**Figure 1.** Short-term senolytic interventions rescue irradiation-induced accelerated ageing. (**A**) Layout of the experiment. (**B**) Frailty index (FI) vs mouse age for non-irradiated (Non-IR, black), irradiated (IR, red), and irradiated plus treated with either D+Q (green) or Navitoclax (blue) mice. Irradiation and treatment times are indicated by vertical lines. Dots indicate FI for individual mice, regression lines and 95% confidence intervals are indicated by bold and dotted lines, respectively. Individual frailty scores are enlarged for clarity in *Figure 1—figure supplement 1*. (**C**) Wire hanging test results (%

*Figure 1 continued on next page*

*Figure 1 continued*

success) under the indicated treatments and ages. (**D**) Maximum speed achieved on rotarod under the indicated treatments and ages. (**E**) Short-term memory assessed as spontaneous alternation in a Y maze under the indicated treatments. (**F**) Alanine transaminase (ALT) activity in plasma at 16 months. (**G**) Tumour prevalence at death. Data are from 12 mice per group at start with attrition to 8–10 mice over the course of the experiment.

The online version of this article includes the following figure supplement(s) for figure 1:

**Figure supplement 1.** Enlarged for clarity (**A-D**).

**Figure supplement 2.** Components of the frailty index (FI).

**Figure supplement 3.** Impact of interventions on epidermal thickness.

**Figure supplement 4.** Validation of senescence markers in liver.

**Figure supplement 5.** Impact of early senolytics treatment on hepatocyte senescence at late age (16 months).

**Figure supplement 6.** Senescence and neuroinflammation markers in hippocampus after early senolytic intervention.

**Figure supplement 7.** Functional and senescence markers in skeletal muscle after early intervention with D+Q or Navitoclax.

**Figure supplement 8.** Cytokine/chemokine concentrations in serum of mice at 16 months of age as measured by cytokine array.

scale, body condition, breathing rate, eye discharge/swelling, whisker loss, and body weight loss. In addition, Navitoclax treatment also reduced loss of fur colour and tumour incidence (*Figure 1—figure supplement 2*). Irradiated mice progressively lost neuromuscular co-ordination as indicated by increasingly poorer performance in the wire hanging (*Figure 1C*) and rotarod (*Figure 1D*) tests. Improvements for both interventions became greater with increasing age, with Navitoclax being more effective than D+Q (*Figure 1C and D*). Short-term memory was assessed using spontaneous alternation frequency in a Y-maze test at 16 months of age (*Fielder et al., 2019*). It was improved substantially following Navitoclax treatment, with a modest improvement following D+Q (*Figure 1E*). Liver damage was assessed by the activity of alanine transaminase (ALT) and aspartate aminotransferase (AST) in serum, which indicates leakage from hepatocytes, at 16 months of age. ALT activity in serum was reduced under both interventions (*Figure 1F*), suggesting that they enabled better liver maintenance. However, intervention-dependent changes were not significant for AST although ALT and AST activities were positively correlated amongst all mice (correlation coefficient = 0.533, p=0.0000234). Picrosirius red staining did not show a significant effect of senolytics on liver fibrosis. Although cohort sizes were not powered to assess long-term survival effects, both treatments tended to result in improved survival and lower tumour prevalence at death in comparison to irradiated control mice (*Figure 1G*). While normal skin ageing is characterised by epidermal thinning, irradiation-induced premature ageing is accompanied by hypertrophy of the epidermis in the skin (*Figure 1—figure supplement 3A,B*). Neither D+Q nor Navitoclax treatment reduced epidermal hypertrophy at late age (*Figure 1—figure supplement 3C*).

Sublethal irradiation resulted in persistently enhanced levels of markers for cellular senescence in multiple tissues of mice for up to 12 months (*Le et al., 2010*; *Seol et al., 2012*). Senescence marker levels in 1 year old irradiated mice were similar to those found in normally ageing mice older than 24 months (*Hudgins et al., 2018*; *Jurk et al., 2014*). We focussed first on liver in which hepatocyte senescence contributes causally to age-associated functional decline (*Jurk et al., 2014*; *Ogrodnik et al., 2017*). As senescence markers, we measured nuclear size and karyomegaly (*Aravinthan and Alexander, 2016*; *Ogrodnik et al., 2017*), nuclear HMGB1 exclusion (*Davalos et al., 2013*), nuclear accumulation of telomere-associated DNA damage foci (TAF), and frequencies of TAF-positive hepatocyte nuclei (*Hewitt et al., 2012*; *Ogrodnik et al., 2017*). Among TAF-positive cells, we assayed both cells with any TAF and those with at least three TAF, because of previous evidence suggesting that the latter might be more representative of cells in 'full' or 'late' senescence (*Zou et al., 2004*; *Jurk et al., 2014*). The markers indicated higher frequencies of senescent hepatocytes at 7 months after irradiation (*Figure 1—figure supplement 4A-F*). Marker changes were similar to those seen in normally ageing mice at ages above 30 months (*Figure 1—figure supplement 4G-K*). Navitoclax intervention at 6 months of age resulted in reduced senescent hepatocyte frequencies in liver 10 months later as indicated by all markers tested (*Figure 1—figure supplement 5A-F*). However, D+Q intervention led to reduced nuclear size and hepatocyte karyomegaly but did not maintain a significant long-term reduction of HMGB1-negative or TAF-positive hepatocytes (*Figure 1—figure supplement 5A-F*).

Senescent cell burden, specifically in the hippocampus, is associated with memory deficits in ageing mice (*Fielder et al., 2020*; *Musi et al., 2018*; *Ogrodnik et al., 2021*). To assess the mechanistic basis for the observed improvements of short-term memory (*Figure 1E*), we measured markers for a senescence-like phenotype (nuclear size, loss of nuclear Lamin B1 expression, and TAF frequencies) and for neuroinflammation (ionized calcium-binding adapter molecule 1 [Iba1]-positive cell density and soma size) in the CA1 and CA3 hippocampal regions (*Figure 1—figure supplement 6*). Both D+Q and Navitoclax reduced nuclear size (*Figure 1—figure supplement 6A,B*) and TAF frequencies (*Figure 1—figure supplement 6E-I*) as markers for a senescent phenotype in CA1 and CA3 pyramidal layer neurons, but Navitoclax had no effect on laminB1 expression (*Figure 1—figure supplement 6C,D*). Similarly, Navitoclax had no effect on frequencies of Iba1-positive microglia (*Figure 1—figure supplement 6J,K*), although microglia soma size was reduced in CA1 and CA3 after both treatments (*Figure 1—figure supplement 6L,M*).

We also examined the effects of D+Q or Navitoclax treatment on morphological and functional parameters of hind limb muscle (*Figure 1—figure supplement 7*). Interestingly, improvements of neuromuscular co-ordination (*Figure 1C*) and strength/endurance (*Figure 1D*) were not associated with enhanced muscle fibre diametre (*Figure 1—figure supplement 7A,B*), decreased frequency of p21-positive myonuclei (*Figure 1—figure supplement 7C,D*), or decreased frequency of TAF-positive myonuclei (*Figure 1—figure supplement 7E*) as senescence marker. Neither irradiation alone nor combination with either of the senolytic interventions changed hind limb muscle fibrosis or fat accumulation as assessed by Picro-SiriusRed/FastGreen staining.

It is often assumed that the effect of senescent cells onto physiological characteristics is mediated via the SASP, especially the induction of a chronic inflammatory state (*Ogrodnik et al., 2019*). Concentrations of 18 cytokines/chemokines that are part of a typical SASP were measured in serum at the end of the experiment by cytokine array (Eve Technologies). Although short senolytic treatment at 6 months of age was sufficient to reduce senescence markers in liver and brain persistently (*Figure 1—figure supplements 5 and 6*), none of the analysed cytokines was significantly different from controls at late age (*Figure 1—figure supplement 8*).

## Late senolytic interventions block further progression of irradiation-induced premature ageing

So far, our data showed that a short senolytic intervention at an early timepoint can rescue irradiation-induced premature accumulation of senescent cells as well as premature physiological ageing. We next asked the question, whether senolytics could still be effective if mice were treated late after irradiation, when premature ageing was already manifest. Animals were again irradiated at 5 months of age, but senolytic interventions were delayed for 6 months and mice were treated with the senolytics D+Q or Navitoclax (using the same regimen as before) at 11 months of age (*Figure 2A*). At this timepoint, the frailty index in irradiated mice was already significantly above than in sham-irradiated mice (*Figure 2B*). However, similar to early intervention, late senolytic treatment did not reduce the frailty index score, but rescued its further accelerated progression (*Figure 2B*). Late intervention with each of the senolytics improved largely the same components of the frailty index as early senolytic intervention, namely, mouse grimace scale, body condition, breathing rate, and eye discharge/swelling (*Figure 1—figure supplement 2*). Mice that had been treated with senolytics at 11 months of age still showed a tendency for improved results of the hanging wire test at 14 months (*Figure 2C*); however, rotarod performance was not better than in irradiated animals (*Figure 2D*). Late intervention with either senolytic improved short-term memory at late age (*Figure 2E*). Neither liver damage (*Figure 2F*) nor tumour incidence (*Figure 2G*) following late senolytic treatment was significantly reduced. Epidermal thickness was unchanged (*Figure 2—figure supplement 2*).

In agreement with a diminished effect of late treatments on liver damage, there was also less impact on persistent systemic cell senescence as assessed by nuclear size, nuclear HMGB1 expression, and TAF frequencies in liver (*Figure 2—figure supplement 3D-F*); however, treatment with Navitoclax still reduced nuclear size (*Figure 2—figure supplement 3A*), karyomegaly (*Figure 2—figure supplement 3B*), and frequencies of HMGB1-negative hepatocytes (*Figure 2—figure supplement 3C*) suggesting a reduced senescent burden. There was no improvement of hind limb myofibre cross-sectional area but rather a tendency (significant for Navitoclax) to reduce it (*Figure 2—figure supplement 4A,B*). There was no reduction of TAF frequencies in muscle (*Figure 2—figure supplement*

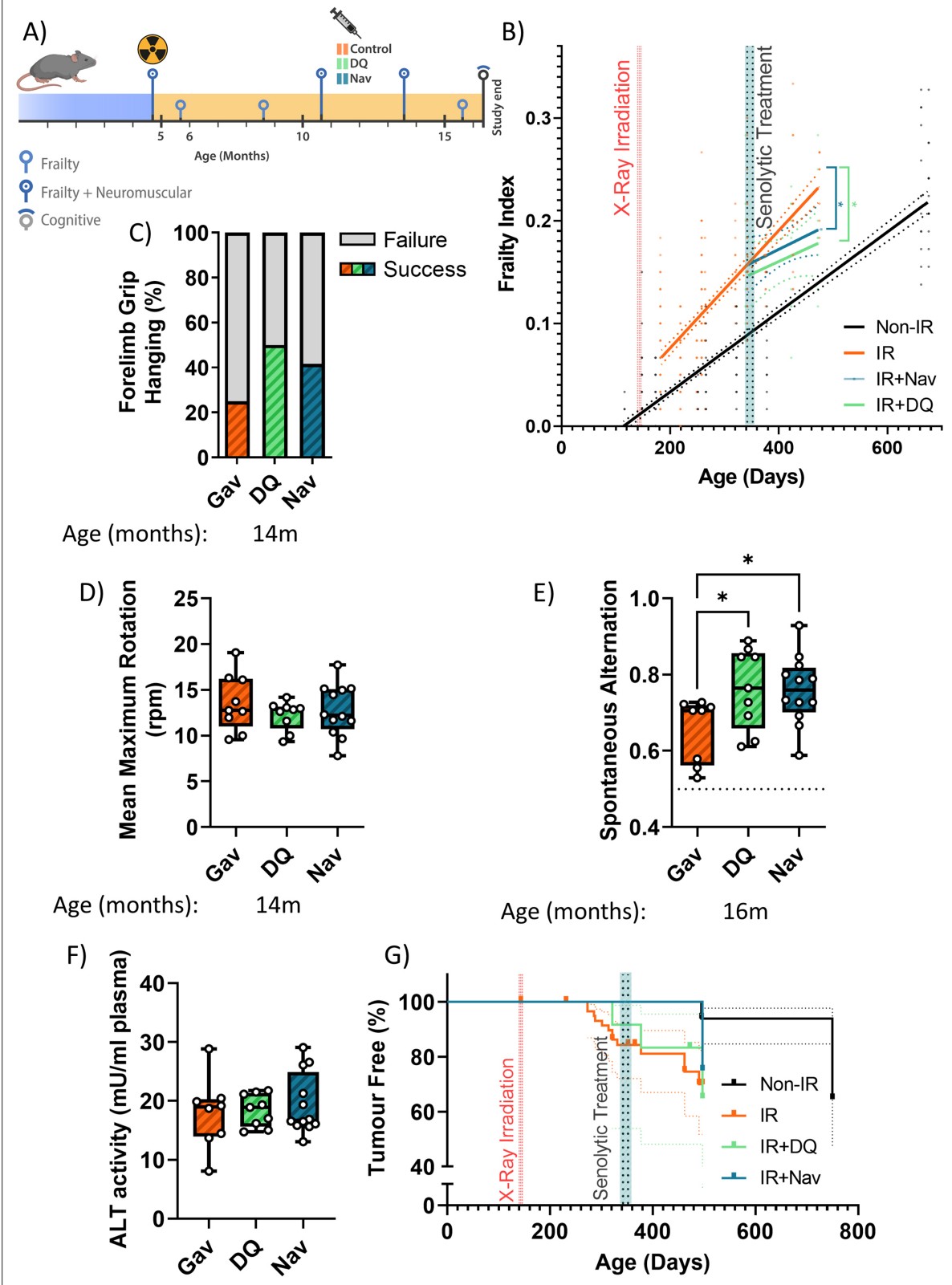

**Figure 2.** Late senolytic interventions partially block further progression of irradiation-induced accelerated ageing. (**A**) Layout of the experiment. (**B**) Frailty index (FI) vs mouse age for non-irradiated (Non-IR, black), irradiated (IR, red), and irradiated plus treated with either D+Q (green) or Navitoclax (blue) mice. Irradiation and treatment times are indicated by vertical lines. Dots indicate FI for individual mice, regression lines and 95% confidence intervals are indicated by bold and dotted lines, respectively. Individual scores are enlarged for clarity in *Figure 2—figure supplement 1*. (**C**) Wire

*Figure 2 continued on next page*

*Figure 2 continued*

hanging test results (% success) under the indicated treatments. (**D**) Maximum speed achieved on rotarod under the indicated treatments at 14 months of age. (**E**) Short-term memory assessed as spontaneous alternation in a Y maze under the indicated treatments. (**F**) Alanine transaminase (ALT) activity in serum, 16 months old. (**G**) Tumour prevalence at death. Data are from 12 mice per group at start with attrition to 8–10 mice over the course of the experiment.

The online version of this article includes the following figure supplement(s) for figure 2:

**Figure supplement 1.** Enlarged for clarity.

**Figure supplement 2.** Late intervention with either D+Q or Navitoclax (at 12 months of age) does not change epithelial thickness at 16 months.

**Figure supplement 3.** Impact of late senolytics treatment on senescence markers in liver.

**Figure supplement 4.** Functional and senescence markers in skeletal muscle after late intervention with D+Q or Navitoclax.

**Figure supplement 5.** Impact of late intervention on neuroinflammation.

*4C*). However, in agreement with improved memory maintenance (*Figure 2E*) we found a reduction of neuroinflammation markers in the CA1 (both markers) and CA3 (microglia soma size only) regions of the hippocampus following both late interventions (*Figure 2—figure supplement 5*).

Together, these data show that a short-term senolytic intervention even if applied at an advanced age still has beneficial effect on irradiation-induced premature progression of frailty and cognitive decline.

## A short-term intervention with the senostatic metformin rescues irradiation-induced premature ageing

Senolytics can have serious side effects, for instance, Navitoclax-induced thrombocytopenia at higher doses (*Demaria, 2017*), that may be limiting for preventive applications. Senostatic or senomorphic caloric restriction mimetics, which block senescence-stabilising signalling, can also reduce net accumulation of senescent cells in tissues (*da Silva et al., 2019*; *Wang et al., 2010*). One example is metformin, which has been shown to act as a senostatic (*Moiseeva et al., 2013*) and has an excellent safety profile as testified by about 70 years of clinical application. We therefore decided to treat our irradiated mice with metformin for a relatively short period (10 weeks), starting at 1 month after irradiation, and assessed the long-term effects of this treatment (*Figure 3A*). The amount of 1 mg/g food, as used here, had previously been found to extend lifespan and healthspan of mice (*Martin-Montalvo et al., 2013*). Similar to senolytic interventions, metformin treatment rescued the enhanced rate of frailty progression due to irradiation (*Figure 3B*). Among the components of frailty, metformin improved mouse grimace scale, body condition, breathing rate, whisker loss, and body weight loss at late age (*Figure 1—figure supplement 2*). It also improved neuromuscular co-ordination as tested by hanging wire test (*Figure 3C*), but had only a minor effect on performance on the rotarod at late age (*Figure 3D*), which might be due to the high body weights of mice fed soaked food. At 16 months of age, metformin-treated animals tended to perform better in the short-term memory test (*Figure 3E*), showed less liver damage (*Figure 3F*) and tumour prevalence at death was reduced to the levels as in sham-irradiated mice (*Figure 3G*). Metformin treatment also tended to reduce irradiation-induced epidermal hypertrophy assessed at 16 months of age (*Figure 3—figure supplement 2*).

All tested senescence markers in livers of metformin-treated mice indicated the reduction of senescent cell frequencies at old age (16 months) (*Figure 3—figure supplement 3A-F*). Similarly, senescence and neuroinflammation markers in the CA1 and CA3 regions of the hippocampus were decreased (*Figure 3—figure supplement 4A-F*). Interestingly, metformin treatment improved skeletal muscle fibre maintenance as shown by larger cross-sectional area of both oxidative and non-oxidative muscle fibres (*Figure 3—figure supplement 5A,B*) and reduced TAF frequencies in myocyte nuclei (*Figure 3—figure supplement 5C*). Metformin treatment also led to a persistent reduction of the levels of the pro-inflammatory SASP components IL-17, CCL2, and TNFα in serum of mice at 16 months of age, e.g., more than half a year after cessation of treatment (*Figure 3—figure supplement 6*).

In conclusion, these data indicate that a relatively short treatment with the senostatic metformin rescues multiple domains of irradiation-induced premature ageing in mice for at least 10 months after cessation of the intervention.

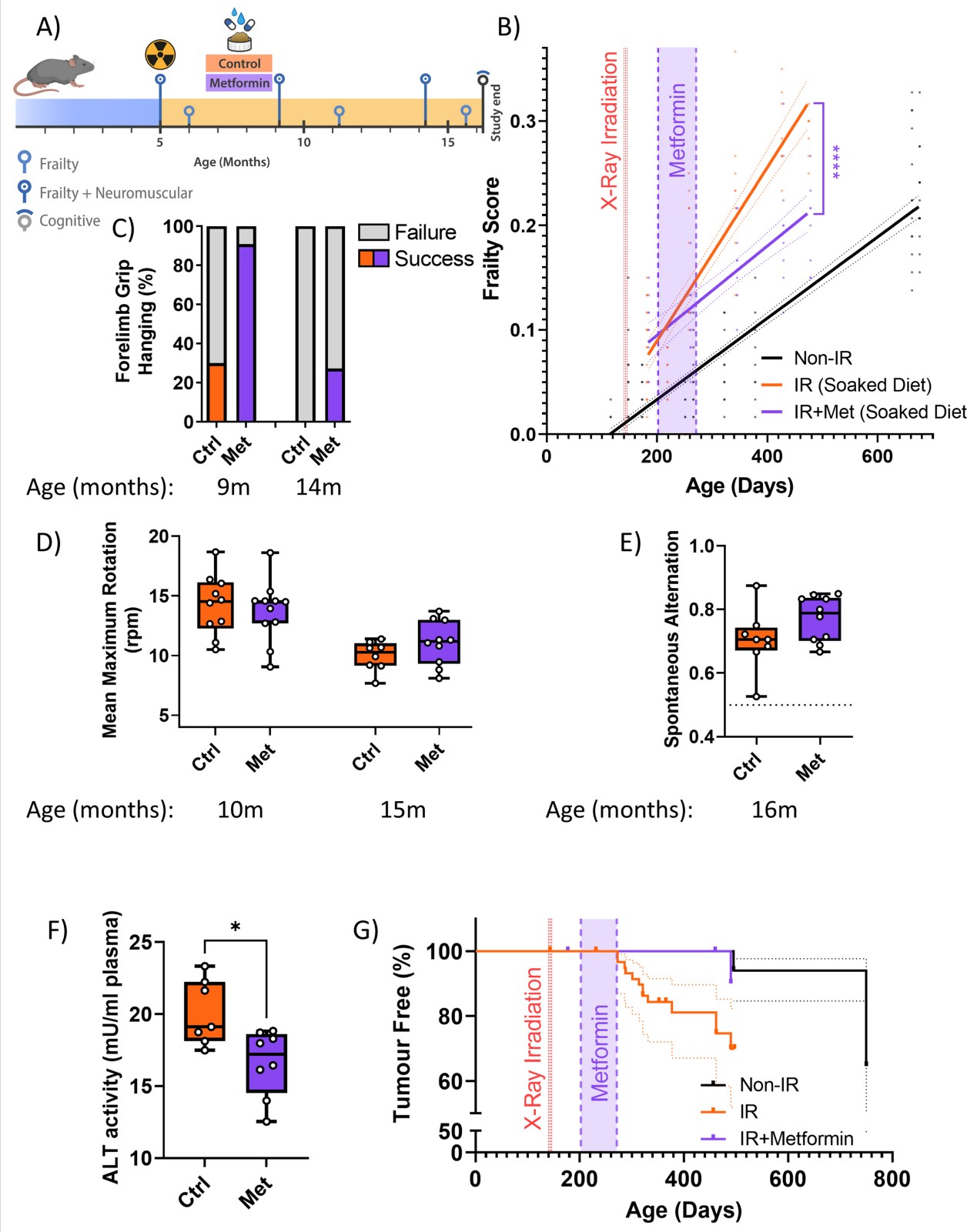

**Figure 3.** A short-term intervention with the senostatic metformin rescues irradiation-induced accelerated ageing. (**A**) Outline of the experiment. Animals were irradiated at 5 months of age and treated with either soaked food (controls) or metformin in soaked food (Met) from 6 months of age for 10 weeks. (**B**) Frailty index (FI) vs mouse age for non-irradiated (no IR, black), irradiated (IR, red), and IR plus treated with metformin (blue) mice. Irradiation and treatment times are indicated by vertical lines. Dots indicate FI for individual mice, regression lines and 95% confidence intervals are

Figure 3 continued

indicated by bold and dotted lines, respectively. Individual scores are enlarged for clarity in *Figure 3—figure supplement 1*. (C) Wire hanging test results (% success) under the indicated treatments and ages. (D) Maximum speed achieved on rotarod under the indicated treatments and ages. (E) Short-term memory assessed as spontaneous alternation in a Y maze under the indicated treatments. (F) Alanine transaminase (ALT) activity in plasma at 16 months of age. (G) Tumour prevalence at death. Data are from 12 mice per group at start with attrition to 8–10 mice over the course of the experiment.

The online version of this article includes the following figure supplement(s) for figure 3:

**Figure supplement 1.** Enlarged for clarity.

**Figure supplement 2.** Treatment of irradiated mice for 2.5 months with metformin (starting at 7 months of age) tends to reduce epidermal thickness at late age (16 months).

**Figure supplement 3.** Impact of metformin on senescence markers in liver.

**Figure supplement 4.** Senescence and neuroinflammation markers in hippocampus after intervention with metformin.

**Figure supplement 5.** Impact of metformin on hind limb muscle.

**Figure supplement 6.** Cytokine/chemokine concentrations in serum of mice at 16 months of age as measured by cytokine array.

## At therapeutic concentrations, metformin inhibits the SASP by reducing NADPH oxidase 4 activity in senescent cells

Metformin can block the SASP, and thus act as a senostatic, by inhibiting complex I of the electron transport chain, thus causing a reduction of mitochondrial ROS production, which in turn will reduce the activity of the NF-κB transcription factor, the major driver of the pro-inflammatory SASP. This pathway has been identified in vitro, using millimolar concentrations of metformin (*Moiseeva et al., 2013*). However, therapeutically achievable metformin concentrations in the vast majority of tissues in mice or man are typically well below 100 μM (*Wilcock and Bailey, 1994*). In permeabilised human fibroblasts in vitro, metformin inhibits complex I-dependent respiration with pyruvate and malate in concentrations around 1 mM or higher but has no detectable effect on oxygen consumption rates (OCR) at 100 μM (*Figure 4A*). Supplementation of the complex II substrate succinate completely restored respiration under metformin, confirming that metformin at high concentrations inhibits complex I specifically (*Figure 4A*). Even when senescent human fibroblasts were treated with various concentrations of metformin for 10 days to mimic longer-term in vivo interventions, low metformin concentrations (100 μM) did not decrease mitochondria-dependent ATP production compared with senescent untreated controls. In contrast, 2 mM metformin shifted cellular ATP production almost entirely to glycolysis with little contribution from mitochondrial oxidative phosphorylation (*Figure 4B*). Interestingly, the senescence-associated enhanced $H_2O_2$ production from whole cells (as measured by Amplex Red assay) was rescued only by low (up to 400 μM), but not by high metformin concentrations (*Figure 4C*). Reduction of ROS production in senescent cells by long-term treatment with low metformin concentrations was confirmed by measuring cellular ROS levels using dihydroethidium (DHE) fluorescence (*Figure 4D*), with a stronger effect for metformin as compared to rapamycin. In accordance with the ROS data (*Figure 4C and D*), low concentrations of metformin were more effective than higher ones in reducing a wider range of cytokines in the secretome of senescent fibroblasts (*Figure 4E*). Together, these data indicate that low, therapeutically relevant concentrations of metformin reduce the release of ROS and SASP cytokines from senescent cells, which can explain the senostatic activity of metformin in vivo. Importantly, this effect was not mediated by complex I inhibition.

To identify potential alternative mechanisms of the senostatic activity of metformin, we subjected human fibroblasts treated with either low (100 μM) or high (2 mM) metformin concentrations to a stress response pathway identifier assay by cytometry by time of flight (CyTOF). Two or three antigens were chosen to represent each of seven cellular stress response pathways, namely, heat shock, oxidative stress response, xenobiotics response, ER-UPR, Mito-UPR, nutrient signalling pathway/autophagy, and DNA damage response (DDR)/senescence, resulting in a panel of 21 antibodies (Table 2). Cells were treated with test interventions for 2 days and analysed by CyTOF using the antibody panel. Starvation, heat shock, and oxidative stress by $H_2O_2$ treatment were used as positive control interventions. In the positive control experiments, activation of heat shock and oxidative stress response pathways was evident following the respective control treatments together with induction of autophagy and a DDR/senescence, while starvation impacted primarily onto the mTOR pathway, together validating

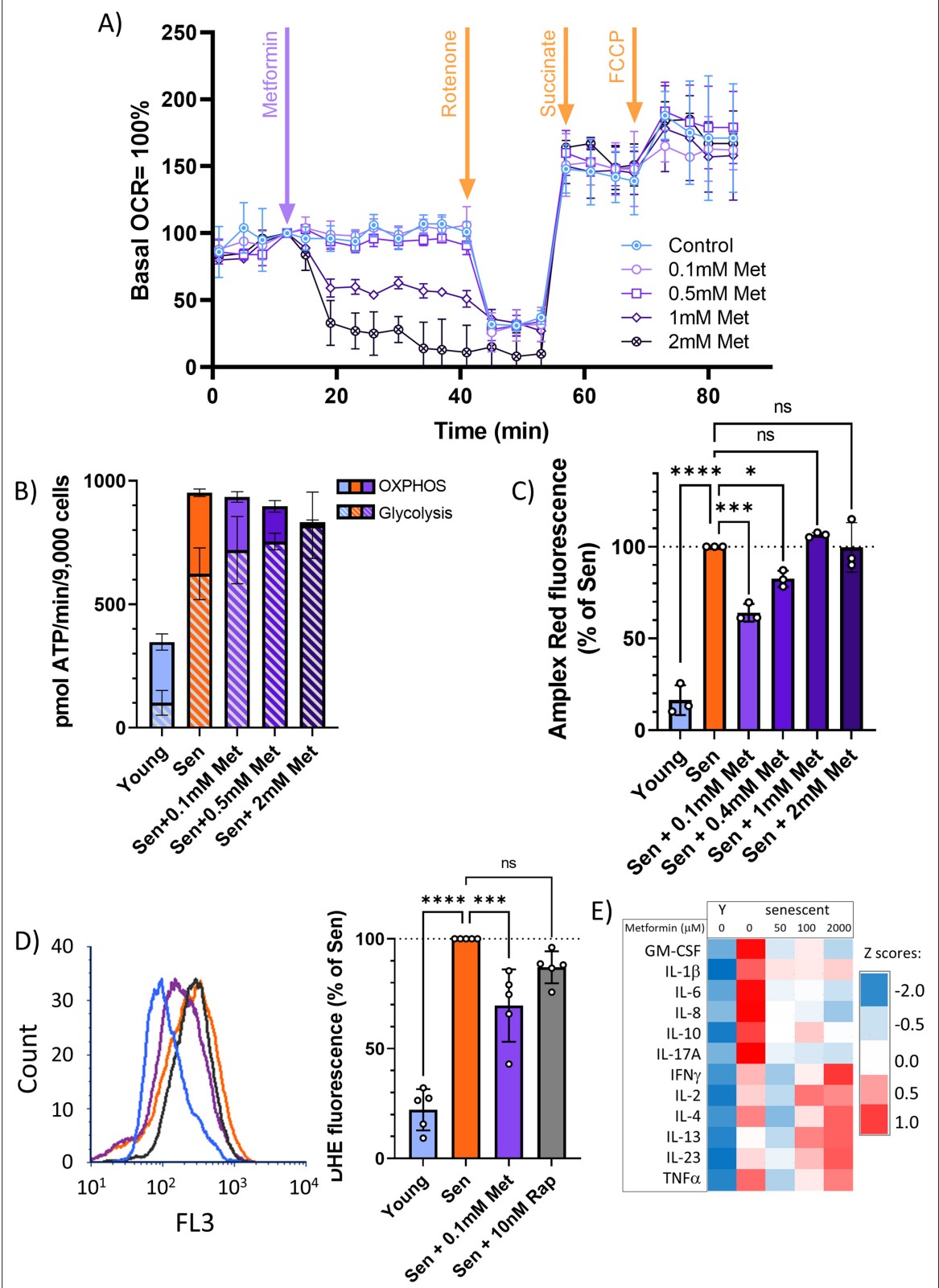

**Figure 4.** At therapeutic concentrations, metformin reduces reactive oxygen species (ROS) and senescence-associated secretory phenotype but not by inhibition of complex I. (**A**) Mitochondrial oxygen consumption rate of permeabilised MRC5 human fibroblasts treated sequentially (at timepoint indicated by arrow) with various concentrations of metformin (Met), 0.5 μM rotenone (Rot), 4 mM succinate (Suc), and 2.5 μM carbonyl cyanide p-trifluoromethoxy-phenylhydrazone. M ± SD, n = 4. (**B**) ATP production rate by oxidative phosphorylation (OXPHOS) and glycolysis in young and

*Figure 4 continued on next page*

Figure 4 continued

senescent MRC5 human fibroblasts treated for 10 days with the indicated metformin concentrations. M ± SD, n = 4. (**C**) Impact of metformin in the indicated concentrations on ROS production measured by AR in human fibroblasts. Cells were induced to senescence by IR and treated with metformin for 10 days. M ± SD, n = 3. (**D**) Impact of rapamycin and metformin on ROS levels in human fibroblasts measured by dihydroethidium (DHE) fluorescence in FACS. Left: representative FL3 histograms. Light blue: young, red: senescent (10d past IR), purple: senescent + 0.1 mM metformin, black: senescent + 10 nM rapamycin. Right: average DHE fluorescence intensities. M ± SD, n = 5. (**E**) Cytokine concentrations in the supernatant of human fibroblasts. N = 2. Senescent fibroblasts were treated with the indicated metformin concentrations for 10 days. Source data are provided as *Figure 4—source data 1*.

The online version of this article includes the following source data for figure 4:

**Source data 1.** Cytokine array primary data.

the assay (*Figure 5A*). Treatment with high metformin reduced the levels of marker proteins in a wide range of pathways, including heat shock, oxidative stress response, ER-UPR, Mito-UPR, nutrient signalling pathway/autophagy, and DDR/senescence. In contrast, low metformin only reduced indicators of oxidative stress response as expected (compare *Figure 4C and D*) and nutrient signalling (*Figure 5A*). This suggested that low metformin might activate autophagy and mitophagy via suppression of the mTOR pathway and thus contribute to a reduction of mitochondrial ROS production in senescence. Therefore, we next examined mitophagy activity in human fibroblasts transfected with the mitophagy reporter mt-mKeima (*Katayama et al., 2011*), which localises to mitochondria and displays a shift in fluorescence emission under low pH, when mitochondria are delivered into lysosomes (indicated in red, *Figure 5B*). As shown before (*Dalle Pezze et al., 2014*; *Korolchuk et al., 2017*), mitophagy activity was reduced in senescent cells. This reduction occurred within hours after irradiation and mitophagy remained low in irradiated cells for multiple days (*Figure 5C*). Treatment with rapamycin improved mitophagy at all timepoints, but 100 µM metformin had no effect (*Figure 5C*). Mitochondrial dysfunction in senescence is characterised by high ROS production together with low respiratory coupling (*Passos et al., 2007*). In accordance with their effects on mitophagy, rapamycin, but not metformin, suppressed senescence-associated mitochondrial superoxide production as measured by MitoSOX fluorescence (*Figure 5D*). Moreover, metformin did not rescue mitochondrial dysfunction in senescent cells as assessed by respiratory control ratio (RCR) with the complex I-linked substrate, pyruvate + malate (*Figure 5E*).

Together, these data suggested that low metformin has no effect on mitochondrial (dys)function in senescence. Therefore, we tested the alternative possibility that it might primarily reduce non-mitochondrial, rather than mitochondrial, ROS production in senescent cells. ROS production by the NADPH oxidase 4 (NOX4) has been shown to contribute to replicative (*Lener et al., 2009*), oncogene-induced (*Weyemi et al., 2012*), and stress-induced senescence (*Goy et al., 2014*) although its knock out had no impact on lifespan in mice (*Rezende et al., 2017*). To test the hypothesis that low metformin might act as a senostatic via reduction of the major cytoplasmic ROS generator NOX4, we first measured the abundance of NOX4 in senescent fibroblasts, which was enhanced as expected (*Figure 6A and B*). Moreover, NOX4 did not colocalise with mitochondria (*Figure 6—figure supplement 1*). A low concentration of metformin (100 µM) reduced NOX4 protein levels in senescent human fibroblasts as shown by both immunofluorescence (*Figure 6A*) and Western blotting (*Figure 6B*). To test whether manipulation of NOX4 alone would be sufficient to explain the senostatic activity of metformin, we overexpressed NOX4 in young fibroblasts and assessed its effects on markers of senescence, SASP, and ROS. In comparison to EGFP-overexpressing controls, NOX4-overexpressing fibroblasts were more often positive for Sen-β-Gal (*Figure 6C*) and produced higher levels of ROS (*Figure 6D*). Importantly, cells overexpressing NOX4 produced significantly more of the SASP interleukin IL-6 (*Figure 6E*), and there was a strong positive correlation between NOX4 and IL-6 levels (*Figure 6F*), but not between EGFP and IL-6 (*Figure 6G*). Finally, we treated fibroblasts in stress-induced senescence with the NADPH oxidase inhibitor diphenyleneiodonium chloride (DPI), which reduced both Sen-β-Gal activity as a marker for the senescent phenotype (*Figure 6H*) and DHE fluorescence, indicative of decreased production of senescence-associated ROS (*Figure 6I*).

Together, our data indicate that metformin at low, therapeutically achievable concentrations reduces senescence-associated ROS production by diminishing NOX4 abundance in senescent cells, which in turn causes the reduction of other facets of the senescent phenotype, importantly including a reduction of SASP production.

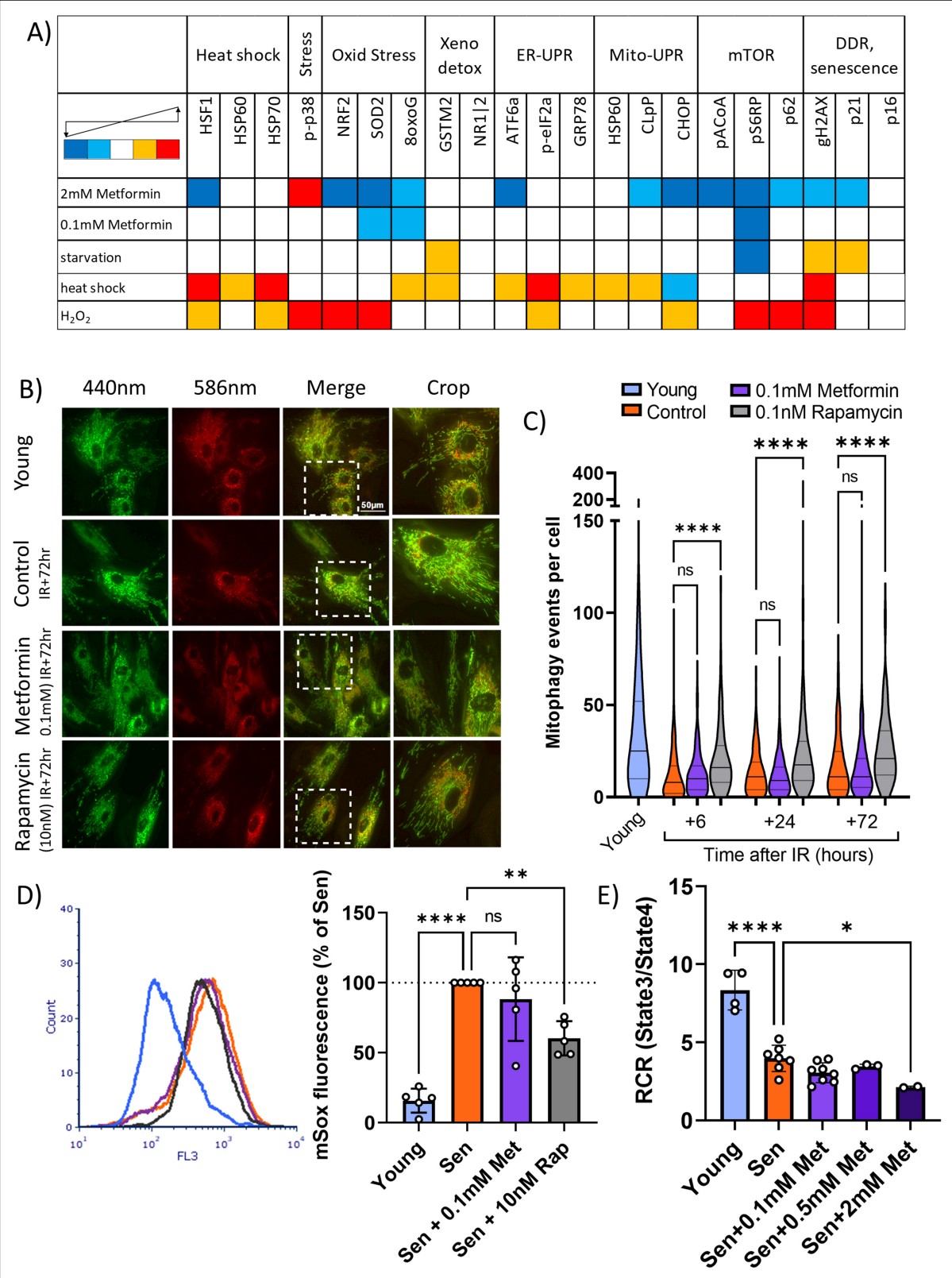

**Figure 5.** Low concentrations of metformin do not improve mitochondrial turnover and function. (**A**) Impact of high (2 mM) and low (100 μM) concentrations of metformin on stress response pathways in human fibroblasts. cytometry by time of flight with the indicated antibodies was performed on human MRC5 fibroblasts treated for 2 days with the indicated interventions. Heat map colour codes indicate strong decrease (dark blue), mild decrease (light blue), no change (white), light increase (amber), or strong increase (red) as exemplified in *Figure 5—figure supplement 1*. Data are

*Figure 5 continued on next page*

*Figure 5 continued*

pooled from two independent experiments. (**B**) Dermal fibroblasts expressing mt-mKeima were irradiated with 20 Gy and treated with either 100 µM metformin or 10 nM rapamycin for 3 days. Red fluorescence indicates mitochondria engulfed in lysosomes. (**C**) Number of mitophagy events per cell. Dermal fibroblasts expressing mt-mKeima were irradiated with 20 Gy and treated with either 100 µM metformin or 10 nM rapamycin for the indicated times. N = 196–271 cells per condition pooled from three biological repeats. (**D**) Impact of rapamycin and metformin on mitochondrial superoxide levels in human fibroblasts either young or at 10 days after IR measured (Sen) by MitoSOX fluorescence in FACS. N = 5. (**E**) Respiratory control ratio (RCR) of mitochondria in fibroblasts at 10 days after IR treated with the indicated concentrations of metformin. N ≥ 3.

The online version of this article includes the following figure supplement(s) for figure 5:

**Figure supplement 1.** Representative histogram overlay examples for use of cytometry by time of flight as stress pathway identifier.

## Discussion

Irradiation is a mainstay of successful therapy for the vast majority of cancers. However, sublethal irradiation causes progressive, premature frailty in both humans (*Ness et al., 2018*; *Robison and Hudson, 2014*) and mice (*Fielder et al., 2019*). Frailty is a medical syndrome characterised by system-wide decreased physiological reserves and thus increased vulnerability (*Rockwood et al., 2015*) predicting multimorbidity and mortality in humans (*Kojima et al., 2018*) and mice (*Whitehead et al., 2014*). In long-term tumour survivors, frailty prevalence reaches a level equal to the general population of 60–70 year olds about 30 years earlier (*Ness et al., 2013*), and in mice frailty progression after sublethal whole-body irradiation is about twice as fast as in non-irradiated animals (*Fielder et al., 2019*). Together with multimorbidity and increased mortality, greatly enhanced frailty is a major component of a serious premature ageing phenotype in long-term tumour survivors, for which no treatment is available so far.

Adjuvant senolytic intervention can relieve some of the consequences of experimental irradiation or chemotherapy in mice. For instance, pharmacogenetic or pharmacologic senolytic intervention up to 12 weeks after irradiation or treatment with the radiation mimetic doxorubicin (partially) corrected treatment-induced loss of immune function (*Chang et al., 2016*; *Palacio et al., 2019*), bone loss (*Chandra et al., 2020*), cardiac dysfunction, and loss of physical activity (*Demaria et al., 2017*), and liver damage (*Baar et al., 2017*). However, whether an adjuvant senolytic intervention would be able to rescue organism-wide radiation-induced premature ageing as documented by frailty levels has not been shown before. Moreover, there is very little data addressing the possible persistence of beneficial effects of senolytic interventions; most published experiments test the outcomes of senolytic treatments only within days to weeks after the intervention. We are only aware of a single paper (*Zhu et al., 2015*) showing that a beneficial effect of a treatment with D+Q (improvement of muscle strength in an irradiated mouse leg) could last for up to 7 months.

Our core hypothesis was that therapy-induced senescence would greatly and persistently accelerate the accumulation of senescent cells by enhancing secondary senescence via bystander signalling, thus causing progressive worsening of ageing-associated symptoms with time following a single bout of DNA-damaging therapies (*Short et al., 2019*). If this hypothesis is correct, eliminating therapy-induced senescent cells by a one-off, short senolytic or senostatic intervention adjuvant to radiation or chemotherapy should be sufficient to prevent progressive premature ageing and to normalise the rate of frailty progression.

Our data support this hypothesis. A single, relatively short adjuvant intervention with either senolytic or the senostatic metformin rescued the radiation-induced accelerated progression of multisystem frailty for at least almost 1 year. We started the interventions at 1 month after completion of radiation, i.e., when signs of acute radiation sickness in the mice had abated but levels of radiation-induced senescence in many tissues were still not significantly above controls (*Mylonas et al., 2021*; *Palacio et al., 2019*; *Palacio et al., 2016*). That this was sufficient to cause a significant reduction of senescence markers in tissues like liver and brain 10 months later is again in agreement with a central role of bystander-mediated accelerated progression of senescence following irradiation.

When senolytic interventions were performed only after enhanced frailty was established, beneficial long-term effects were reduced. There was no longer an effect on muscular or liver function and little improvement on senescence markers in liver at old age. Similarly, *Mylonas et al., 2021* recently reported that a late intervention with Navitoclax reduced senescence markers in kidney but did not improve kidney fibrosis. However, late interventions were still efficient in rescuing further

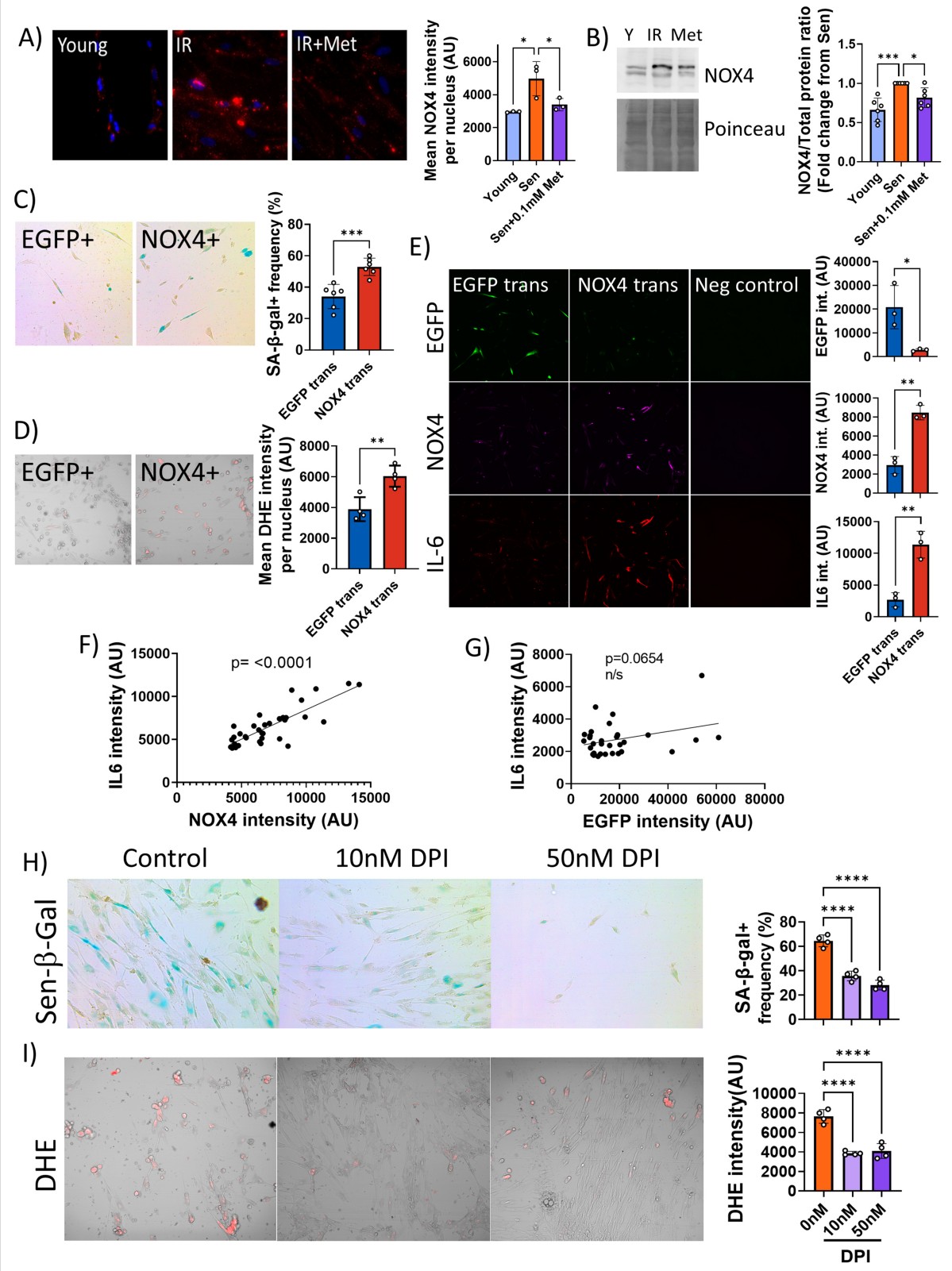

**Figure 6.** Low concentrations of metformin reduce reactive oxygen species in senescence via suppression of NOX4. (**A**) Human fibroblasts were irradiated with 20 Gy and treated with 100 µM metformin for 10 days. Left: Representative images of cells treated as indicated, red: NADPH oxidase 4 (NOX4) immunofluorescence, blue: DAPI. Right: Quantification of NOX4 fluorescence intensity. (**B**) Left: Representative NOX4 Western blot. Cells treated as above. A complete blot is provided as *Figure 6—source data 1*. Right: Average NOX4 signal intensity, normalised to total protein. (**C**) Left:

*Figure 6 continued on next page*

*Figure 6 continued*

Sen-β-Gal staining in EGFP- and NOX4-overexpressing fibroblasts. Right: Average frequencies of Sen-β-Gal-positive cells. (**D**) Left: Dihydroethidium (DHE) staining in EGFP- and NOX4-overexpressing fibroblasts. DHE fluorescence in red, cells visualised in phase contrast. Right: Average DHE fluorescence intensity per cell. (**E**) Co-staining for NOX4 (red) and IL-6 (green) on EGFP- or NOX4-transfected fibroblasts. Right: Fluorescence intensity levels for EGFP (top), NOX4 (middle), and IL-6 in EGFP- and NOX4-transfected cells. (**F**) Correlation between cellular NOX4 and IL6 fluorescence signals in NOX4-transfected cells. (**G**) Correlation between cellular EGFP and IL6 fluorescence signals in EGFP-transfected cells. (**H**) Left: Sen-β-Gal staining in senescent fibroblasts treated with the indicated concentrations of diphenyleneiodonium chloride (DPI). Right: Frequencies of Sen-β-Gal-positive cells. (**I**) Left: DHE staining in senescent fibroblasts treated with the indicated concentrations of DPI. DHE fluorescence in red, cells visualised in phase contrast. Right: Average intensity of DHI fluorescence per cell. All experiments N ≥3.

The online version of this article includes the following source data and figure supplement(s) for figure 6:

**Source data 1.** Full-length NOX4 Western Blot.

**Figure supplement 1.** NADPH oxidase 4 (NOX4) does not colocalise with mitochondria.

frailty progression and liver maintenance and tended to improve short-term memory at high age consistent with a significant reduction of markers for neuroinflammation. This confirms recent data (*Yabluchanskiy et al., 2020*) showing improvements of brain senescence markers, neurovascular coupling, and memory after relatively late (3 months past whole-brain irradiation) pharmacogenetic (ganciclovir in 3MR mice) or Navitoclax intervention. Our results might be interpreted to suggest that ongoing brain cell senescence, including neuron senescence, could be a physiologically relevant driver of neuroinflammation.

While senolytic drugs work in principle on a systemic level, their senolytic activity is cell type-specific. For instance, Navitoclax has senolytic activity against HUVECs and fibroblasts, but not adipocytes in vitro (*Zhu et al., 2016*), while D+Q in combination eliminated senescent HUVECs, fibroblasts, and fat progenitor cells (*Zhu et al., 2015*). The specific activity of these senolytics against many other senescent cell types is not known. Our data show cell type-specific differences in the capacity of Navitoclax vs D+Q for long-term reduction of senescence markers in vivo: under an early intervention regime, Navitoclax and D+Q reduced hippocampal pyramidal neuron senescence at late age equally well but had no effect on senescence marker in muscle fibres, while senescence markers in liver hepatocytes were only efficiently reduced by Navitoclax but not by D+Q. This pattern was different from the physiological responses in the same tissues: Navitoclax and D+Q improved function in the liver (as measured by ALT release), brain (short-term memory), and muscle (rotarod and hanging wire performance) with only a slightly better performance of Navitoclax in terms of memory and muscle function. The senostatic metformin might be expected to have less cell type specificity, and in fact it did reduce late age senescent cell frequencies both in liver, brain, and muscle. Together, these data suggest that cognitive improvement by senolytics could be tissue-autonomous, but that there must be significant contributions from systemic effects causing improvements in muscular and possibly liver function. However, measuring the serum abundance of 18 major interleukins at late age, we did not find evidence for obvious impacts of any of the interventions, suggesting that the mediation of systemic effects might be more complex than persistent suppression of some pro-inflammatory cytokines.

The capacity of the dietary restriction mimetic metformin to extend lifespan and healthspan in mice (*Martin-Montalvo et al., 2013*) and probably man (*Campbell et al., 2017*; *Wang et al., 2017*) is well documented. Among other mechanisms, metformin, like dietary restriction itself (*da Silva et al., 2019*), inhibits the pro-inflammatory SASP (*Moiseeva et al., 2013*) and thus limits the spread of senescence via bystander effects. We therefore expected it to be similarly effective as senolytics in reducing persistent therapy-induced senescence and its physiological consequences, and our results confirmed this expectation.

It has been shown that metformin inhibits complex I of the mitochondrial electron transport chain, thus reducing ROS production (*Moiseeva et al., 2013*). ROS levels are high in senescent cells (*Passos et al., 2010*; *Passos et al., 2007*) and their ability to activate the SASP is well documented (*Coppé et al., 2010*; *Nelson et al., 2018*). Thus, it has been suggested that metformin diminishes the SASP via complex I inhibition (*Moiseeva et al., 2013*). However, metformin inhibits complex I only at supraphysiological concentrations. At therapeutically achievable and effective concentrations, it still reduces cellular ROS (*Figure 4C and D*) and the SASP (*Figure 4E*), but has no effect on mitochondrial complex I-linked oxygen consumption (*Figure 4A and B*).

Complex I is a multiprotein enzyme consisting of more than 40 subunits. In ageing and cellular senescence, incompletely assembled subcomplexes of complex I accumulate and contribute to cellular ROS production (*Miwa et al., 2014*). Decreased efficacy of mitophagy in senescence and ageing is a possible cause of this accumulation (*Dalle Pezze et al., 2014*; *Korolchuk et al., 2017*). Therefore, we next tested whether metformin at therapeutic concentrations might be able to rescue the low mitophagy activity in senescence, using rapamycin as a positive control. Surprisingly, low metformin concentrations did not improve mitophagy (*Figure 5B and C*). In agreement with this, they did not reduce mitochondrial ROS production as measured by MitoSOX fluorescence (*Figure 5D*) and did not rescue low mitochondrial coupling (*Figure 5E*) in senescence, suggesting a non-mitochondrial pathway for reduction of ROS by therapeutic metformin concentrations. Suppression of NOX4 was identified as a mechanism by which low metformin reduces senescence-associated ROS production and SASP (*Figure 6*). It had been shown that metformin suppresses NOX4 induction via activation of AMPK both at concentrations as high as 10 mM (*Sato et al., 2016*) and as low as 0.1 mM (*Shi and Hou, 2021*). However, protective effects of metformin against the radiation mimetic doxorubicin were only observed at low concentrations due to the suppression of platelet-derived growth factor receptor (PDGFR) expression by high metformin (*Kobashigawa et al., 2014*). In vivo, high metformin doses shortened the lifespan of mice (*Palliyaguru et al., 2020*). Other receptor tyrosine kinases including FGFR1 have also been involved in the cellular responses to metformin (*Shi et al., 2021*). In low concentrations, metformin was able to suppress oxidative stress-associated senescence of adipose-derived stromal cells via activation of AMPK, but it is unclear whether this was NOX-dependent (*Le Pelletier et al., 2021*). A more detailed analysis of the network of signalling pathways mediating NOX4 suppression by low metformin is clearly warranted but remains outside the scope of the present study.

Our study has a number of limitations. Whole-body irradiation is an oversimplified model for therapeutic irradiation of tumour patients. However, experiments with localised radiation to the brain have also shown improvements in cognition immediately after senolytic intervention (*Yabluchanskiy et al., 2020*). Moreover, senolytic intervention with D+Q after irradiation targeted to a single leg resulted in long-term improvements of muscle function (*Zhu et al., 2015*). These, together with our results, strongly suggest that senolytic or senostatic interventions could be effective against progressive frailty, multimorbidity, and mortality even in realistic radiation therapy situations. We have now established a more realistic mouse model for targeted brain tumour radiation therapy. Pilot phenotyping results show progressive frailty development together with cognitive decline not dissimilar to the effects of whole-body irradiation, and results of an intervention study will be published as soon as possible.

Ageing as a trait shows strong sexual dimorphism in many species including mice (*Bronikowski et al., 2022*) and man (*Hägg and Jylhävä, 2021*). Unsurprisingly, both lifespan- (*Nadon et al., 2017*) and healthspan-extending interventions (*Selman et al., 2009*) frequently have sex-dependent efficacies. It is therefore important to test the impact of senolytic/senostatic interventions post-irradiation in both males and females. This could not be done in the present cohorts due to funding restrictions but will be done in the near future.

Furthermore, we studied mice that were irradiated as young adults, and our results would thus be most directly transferable to adolescent tumour survivors. However, frailty, cognitive decline, and multimorbidity are specifically important problems in childhood tumour survivors. Moreover, relatively few comprehensive data are available about premature ageing phenotypes in the largest patient group, those who developed tumours at old age. Severity of radiation effects and efficacy of interventions might well be different in both the very young and the very old. Studies over a wide age range in mice should be performed in the future, using more realistic therapy models.

In conclusion, we have shown that short senolytic or senostatic interventions can effectively rescue premature progressive frailty and accelerated ageing induced by whole-body irradiation over a significant part of the life history in male mice. We believe these results warrant further efforts to translate senolytic and senostatic interventions towards an adjuvant therapy for long-term tumour survivors.

# Materials and methods

**Key resources table**

| Reagent type (species) or resource | Designation | Source or reference | Identifiers | Additional information |
|---|---|---|---|---|
| Strain, strain background (*Mus musculus*, male) | Wild-type | Charles River | C57BL/6 | |
| Cell line (*Homo sapiens*) | Human foetal lung, male | ECACC | MRC5, Cat-Nr. 05072101 | |
| Antibody | Anti-HMGB1 (Rabbit monoclonal) | Abcam | Cat# ab79823, RRID:AB_1603373 | (1:250) |
| Antibody | Anti-TOMM20 (Mouse monoclonal) | Abcam | Cat# ab56783, RRID:AB_945896 | (1:200) |
| Antibody | Anti-Iba-1 (Rabbit monoclonal) | Abcam | Cat# ab178846, RRID:AB_2636859 | (1:2000) |
| Antibody | Anti-LMNB1 (Rabbit polyclonal) | Abcam | Cat# ab16048, RRID:AB_443298 | (1:200) |
| Antibody | Anti-NOX4 (Rabbit monoclonal) | Abcam | Cat# ab109225, RRID:AB_10861375 | IF (1:200) WB (1:2000) |
| Antibody | Anti-IL6 (Mouse monoclonal) | Abcam | Cat# ab9324, RRID:AB_307175 | (1:500) |
| Antibody | Anti-p21 (Rabbit monoclonal) | Abcam | Cat# ab109520, RRID:AB_10860537 | (1:100) |
| Antibody | Anti-Phosphor-Histone H2AX (Rabbit monoclonal) | Cell Signalling Technology | Cat# 9718, RRID:AB_2118009 | (1:250) |
| Antibody | Anti-Beta-Actin (Rabbit monoclonal) | Cell Signalling Technology | Cat# 5125, RRID:AB_1903890 | WB (1:1000) |
| Recombinant DNA reagent | pCHAC-mt-mKeima | Addgene | Cat-Nr. #72,342 | |
| Recombinant DNA reagent | pcDNA3.1-hNOX4 | Addgene | Cat-Nr. #69,352 | |
| Recombinant DNA reagent | pcDNA3.1(+)eGFP | Addgene | Cat-Nr. #129,020 | |
| Commercial assay or kit | Human Cytokine Array Proinflammatory focused 13-plex Assay | Eve Technologies | HDF13 | |
| Commercial assay or kit | Mouse High Sensitivity 18-Plex Discovery Assay | Eve Technologies | MDHSTC18 | |
| Commercial assay or kit | eBioscience Foxp3/Transcription Factor Staining Buffer Set | ThermoFisher | 00-5523-00 | |
| Chemical compound, drug | Dasatinib | Merck | CDS023389 | |
| Chemical compound, drug | Quercetin | Merck | #1592409 | |
| Chemical compound, drug | Navitoclax | Stratech | #285063-USB | |
| Chemical compound, drug | Metformin HCL API | Pharmahispania | | |
| Chemical compound, drug | Polyethylenglycol 400 | Merck | #8074851000 | |

## Study design

This study addressed the hypothesis that premature frailty and accelerated ageing after sublethal irradiation are caused by accelerated accumulation of senescent cells triggered by bystander signalling from radiation-induced senescence, and that thus a short treatment with either a senolytic or the senostatic metformin would be protective. A series of controlled laboratory experiments were performed to determine the progression of frailty and other markers of biological age following irradiation with and without intervention at prespecified time points. All studies were conducted at Newcastle University in agreement with ARRIVE guidelines (*Kilkenny et al., 2010*). On the basis of previous work, two-sided two-sample test was used to generate animal group sizes. Incorporating expected attrition rates during long-term follow-up, this resulted in group sizes of 12 animals at start of the experiments for physiological phenotyping. Data collection was performed at prespecified

time points unless limited by animal distress as identified by facility staff and/or veterinary surgeons. Primary and secondary end points were prespecified. Each mouse represented one experimental unit. Mice were coded with randomised allocation to experimental groups and housing cages. Data collection, tissue collection, and tissue analyses were done by staff members blinded to experimental group allocation with unblinding performed only after data acquisition was complete. For ex vivo assessments, power calculations informed by previous experience indicated a minimum number of five animals per group. Tissues were either randomly selected or all available tissues were used for analysis.

To address the mechanism of senostatic activity of metformin in therapeutically achievable concentration, a stepwise series of controlled laboratory in vitro experiments was performed, where experimental outcomes guided alternative hypotheses. Individual cell culture dishes represented experimental replicates except in the case of mitophagy experiments, where each cell was treated as a biological unit. Again, in vitro treatments were coded, and codes were broken only after data collection and analysis. In vitro experiments were independently reproduced at least three times.

## Animals

Male C57Bl/6 J mice were bought past weaning from Charles River and were maintained in groups of six littermates per cage as described (*Cameron et al., 2012*). The mice were fed standard pelleted food (CRM-P formulation rodent diet, SDS Diets), except those used for metformin treatment and their control, which received soaked food (same as above) with or without metformin from 1-month post-IR. Mice were sacrificed at the end of the study by cervical dislocation and tissues harvested, and stored in 4% paraformaldehyde for 24 hours for paraffin embedding, or frozen in liquid nitrogen. The work was licensed by the UK Home Office (PB048F3A0) and complied with the guiding principles for the care and use of laboratory animals.

## Irradiation

At 5 months of age, mice were sublethally irradiated thrice (NDT320, 225kV) with 3 Gy of X-ray irradiation, with two days of recovery between each dose, as described (*Fielder et al., 2019*). Mice received 1% Baytril solution in drinking water for 2 days before, and for 14 days after, to the start and end of irradiation, respectively.

## Senolytic and senostatic treatments

Mice were orally gavaged with either 5 mg/kg/day dasatinib and 50 mg/kg/day quercetin or with 5 mg/kg/day Navitoclax for 10 days total (5 days, 2 days recovery, and 5 days). Compounds were prepared for oral gavage in 10% polyethylene glycol (PEG400). Control mice were gavaged with PEG400 only. Interventions were started at 1 month post-irradiation for the early intervention group, and 7 months post-irradiation for the late intervention group. Dasatinib (CDS023389), quercetin (1592409), and PEG400 (8074851000) were purchased from Sigma-Aldrich (now Merck). Navitoclax (285063-USB) was purchased from Stratech.

Metformin hydrochloride was a kind gift from FARMHISPANIA, (Barcelona) and prepared at 1 g/kg in dry food (0.1% (w/w) in food) and provided at 6 mg/6 ml water in 6 g food per mouse in the cage as soaked food. The control group also received soaked food without metformin. Treatment was started at 1 month post-irradiation and was given daily for 10 weeks.

## Mouse phenotyping

Frailty was assessed regularly using a 30-parameter index based on *Whitehead et al., 2014*, with modifications as described in *Fielder et al., 2019*. Rotarod, wire hanging and spontaneous alternation Y-Maze were performed as described in *Fielder et al., 2019*. Tumour incidence at death was assessed by gross pathological examination.

## Immuno-histochemistry (IHC) and immunofluorescence

Paraformaldehyde (PFA)-fixed paraffin embedded tissue samples were cut and stained with primary and secondary antibodies as detailed in *Table 1*, see also *Fielder et al., 2020* for a step-by-step protocol. Fixed cells were blocked with 2% normal goat serum +0.1% BSA and stained overnight with the primary antibody at 4 C (*Table 1*).

**Table 1.** Immunostaining and blotting methods.

| Tissue | Thickness | Technique | Primary antibody | Cat No, vendor | Dilution | Secondary antibody | Cat No, vendor | Dilution | Detection |
|---|---|---|---|---|---|---|---|---|---|
| Liver | 3 | IF | Rabbit anti-HMGB1 | Ab79823 (Abcam) | 1:250 | Goat Anti-Rabbit IgG H&L, Texas Red | Ab6719 (Abcam) | 1:500 | |
| | 3 | IF | Mouse anti-TOMM20 | Ab56783 (Abcam) | 1:200 | Goat anti-mouse (Alexa Fluor 594) | Ab150116 | 1:1,000 | |
| Quads | 3 | IF | Rabbit anti-HMGB1 | Ab79823 (Abcam) | 1:250 | Goat anti-rabbit (Alexa Fluor 594) | A32740 (ThermoFisher) | 1:1,000 | |
| | 10 | IHC | Rabbit anti-Iba1 | Ab178846 (Abcam) | 1:2000 | Biotinylated Goat anti-rabbit | BA-1000 (Vector labs) | 1:250 | VECTASTAIN ABC-HRP Kit, NovaRED (Vector labs) |
| | 3 | IF | Rabbit anti-γH2A.X primary antibody | 9,718 (Cell Signalling) | 1:250 | Biotinylated Goat anti-rabbit | BA-1000 (Vector labs) | 1:250 | Fluorescein Avidin DCS (1:500) (Vector labs) |
| Brain | 3 | IF | Rabbit anti-Lamin B1 | ab16048 (Abcam) | 1:200 | Biotinylated Goat anti-rabbit | BA-1000 (Vector labs) | 1:250 | Fluorescein Avidin DCS (1:500) (Vector labs) |
| | | | Anti-NADPH oxidase 4 antibody | ab109225 (Abcam) | 1:200 | Goat Anti-Rabbit IgG H&L (Alexa Fluor 594) | ab150080 (Abcam) | 1:1,000 | |
| | | | Anti-IL6 antibody | ab9324 (Abcam) | 1:500 | Anti-Mouse IgG (H+L) Alexa Fluor 488 | A-11017 (vector labs) | 1:1,000 | |
| | | | Rabbit anti-p21 | ab109520 (Abcam) | 1:100 | Goat Anti-Rabbit IgG H&L (Alexa Fluor 594) | ab150080 (Abcam) | 1:1,000 | |
| MRC5 Cells | ICC | | Rabbit anti-HMGB1 | ab79823 (Abcam) | 1:250 | Goat Anti-Rabbit IgG H&L (Alexa Fluor 594) | ab150080 (Abcam) | 1:1,000 | |
| Protein | WB | | Anti-NADPH oxidase 4 antibody | ab109225 (Abcam) | 1:2000 | Goat Anti-Rabbit IgG H&L (HRP) | ab6721 (Abcam) | 1:10,000 | |
| | | | Anti-β-Actin antibody | 5,125 (Cell Signaling) | 1:1,000 | Goat Anti-Rabbit IgG H&L (HRP) | ab6721 (Abcam) | 1:10,000 | |

## Immuno-fluorescence in situ hybridisation (immuno-FISH)

The immuno-FISH for TAF were performed as previously described (*Hewitt et al., 2012*, *Fielder et al., 2020*) with the following modifications for quadriceps: The blocking step used 1% BSA, 5% Normal Goat Serum in PBS, for 30 min at 30 °C. Fluorescein Avidin DCS was substituted with Texas Red-labelled Avidin DCS (Vector Laboratories) in PBS for 30 min at 30 °C. CCCTAA Cy-3 probe was substituted for TTAGGG probe (Pangene).

## Microscopy and image analysis

IHC images were taken using a widefield light microscope ECLIPSE E800 (Nikon, Japan) at total magnification of 100 x. Microscopy for IF and immuno-FISH was performed using a DMi8 fluorescence microscope (Leica, Germany) with total magnification of 400 x for IF and 630 x (with Z-stack/depth) for immuno-FISH.

Positive and negative nuclei were manually identified by observers blinded to the treatment groups, counted on five images per animal, and the average was calculated as the individual value for the sample/animal. Nuclear size was manually measured with ImageJ software (NIH, USA).

Epidermal thickness was measured on 3 µm back skin sections stained with Picro-sirius red/fast green. Three different regions were imaged per animal, and 25 measurements were taken in each region using the straight-line tool in ImageJ.

To identify TAF in brain, the colocalisation of DNA damage foci and telomeres were detected manually and confirmed in Icy software through 3D image setup as described (*Hewitt et al., 2012*, *Fielder et al., 2020*). For liver and muscle, detection and 3D location of DNA damage foci and telomeres was automated using Icy software (Institut Pasteur & France Bioimaging, France). A Python programme was used to assess their colocalisation. Results from this automated counting were validated against manual counts in individual liver and muscle sections.

## Western blotting

Cells were collected using Accutase (StemCell Technologies #07922) and lysed using RIPA buffer supplemented with protease inhibitors. Western blotting was performed as described (*Miwa et al., 2014*) with antibodies against NOX4 and β-Actin as detailed in *Table 1*.

## Liver function assessment

Liver function was assessed using the Alanine Transaminase Activity Assay Kit (Abcam, ab105134) and Aspartate Aminotransferase Activity Assay Kit (Abcam, ab105135) according to the manufacturer's instructions. Average of duplicates was used as individual data of a sample/animal, and then the data were grouped by treatment types and compared.

## Cell culture

Human lung MRC5 fibroblasts were grown in Dulbecco's modified Eagle's medium (DMEM, Sigma, # D5671) supplemented with 10% heat-inactivated foetal Bovine Serum (FBS, Sigma), 100 units/ml penicillin, 100 µg/ml streptomycin, and 2 mM L-glutamine at 37 °C in a humidified atmosphere with 5% $CO_2$. Stress-induced senescence was induced by X-ray irradiation with 20 Gy or (for DPI experiments) with 200 µM $H_2O_2$ in serum-free medium.

For metformin treatment, medium was replaced with fresh medium containing 100 µM metformin or DMSO (vehicle control) immediately post irradiation. Treatment was maintained for 10 days, with medium changes every 3 days.

For NOX4 overexpression, pcDNA3.1-hNOX4 (Addgene #69352) was used, with pcDNA3.1(+) eGFP (Addgene #129020) as control. Plasmids were extracted using the EndoFree Plasmid Maxi Kit (Qiagen 12362). 80% confluence MRC5 cells (PD 15–25) were transfected using 500 ng of plasmid per well with Lipofectamine 3,000 (ThermoFisher L3000001). For selection, cells were grown for 1 week in G418 (400 µg/ml) from the 3rd day post-transfection and fixed with 4% PFA for staining.

For NOX4 inhibition, senescent MRC5 cells were treated with diphenyleneiodonium chloride (DPI, Bio-Techne 4673-26-1 at either 50 nM, 10 nM, or DMSO control) for 3 days.

For Sen-β-Gal staining, cells were fixed for 5 min with 2% PFA in PBS-Mg before incubation with the staining solution (PBS-Mg containing 1 mg/ml X-gal, 5 mM potassium ferrocyanide, and 5 mM potassium ferricyanide, pH 5.5) overnight at 37 C.

## Mitophagy measurement

Neonatal human dermal fibroblasts (HDFns) were transduced with lentiviruses containing pCHAC-mt-mKeima (Addgene plasmid #72342) (*Lazarou et al., 2015*). Cells were irradiated with 20 Gy X-ray radiation, and the mt-mKeima signal was measured up to 3 days later. During this time, cells were treated with metformin (100 µM) or rapamycin (10 nM). The live-cell mt-mKeima signal was captured on a Leica DMi8 inverted microscope with a 63 x oil objective. Numbers of red dots per cell, indicating lysosomal mt-mKeima signal, were quantified using ImageJ.

## ROS measurements

Cells were stained with dihydroethidium (DHE, ThermoFisher Scientific) to measure intracellular peroxides or with MitoSOX (ThermoFisher Scientific) to assess mitochondrial superoxide. Cells were incubated with either 10 µM DHE or 5 µM MitoSOX for 30 min at 37 °C in the dark and analysed by flow cytometry or in a DMi8 fluorescence microscope (Leica).

Extracellular release of hydrogen peroxide was measured by Amplex Red assay (ThermoFisher Scientific) as described (*Miwa et al., 2016*) in a 96 well plate using a fluorescent plate reader (FLUO-star Omega, BMG Labtech) at excitation 544 nm and emission 590 nm at 37 °C.

## Oxygen consumption rates

Cellular OCR and media acidification rates (extracellular acidification rate, ECAR) in intact cells were measured in parallel using a Seahorse XF24 analyser in unbuffered basic media (DMEM, Sigma, #D5030) supplemented with 5 mM glucose, 2 mM L-Glutamine, and 3% FBS. Whilst the measurements were taken, the following compounds were injected to test mitochondrial activity and cellular bioenergetics: Oligomycin (0.5 µM) to inhibit ATP synthase, carbonyl cyanide p-trifluoromethoxy-phenylhydrazone (FCCP) (2.5 µM), a respiratory chain uncoupler, 2-deoxyglucose (2DG) (80 mM), a glucose analogue competitively inhibiting glucose uptake and glycolytic flux, and Rotenone (0.5 µM) and Antimycin (2.5 µM), mitochondrial complex I and complex III inhibitors, respectively. Data analysis to calculate absolute ATP production rates was carried out using the methods described by *Mookerjee and Brand, 2015* taking into account the acidification rates due to mitochondrial $CO_2$ production (*Birket et al., 2011*; *Brand, 2005*).

Permeabilised cells were used to measure mitochondrial OCR using Pyruvate (10 mM) and Malate (1 mM) as complex I-linked substrate. Cells were permeabilised using plasma membrane permeabiliser (PMP, Agilent Technologies) according to manufacturer's instructions, and oxygen consumption was measured in medium containing 220 mM Mannitol, 70 mM Sucrose, 10 mM $KH_2PO_4$, 5 mM $MgCl_2$, 2 mM Hepes, 1 mM EGTA, and 0.2% (w/v) Fatty Acid Free BSA. To determine the effects of metformin on mitochondrial complex I activity, sequential additions of metformin (at concentrations as indicated in *Figure 4A*), Rotenone (0.5 µM), Succinate (4 mM), and FCCP (4 µM) were made. For determination of RCR, permeabilised cells respiring with Pyruvate (10 mM) and Malate (1 mM) received 4 mM ATP (State 3) followed by 0.5 µM Oligomycin (State 4). RCR was calculated as state 3 divided by state 4 respiration rates.

## Cytokine measurement

Cytokines secreted from MRC5 cells were analysed by Human Cytokine Array Proinflammatory focussed 13-plex Assay (Eve Technologies, Calgary, Canada). The cells were grown in 75 cm flasks, and the culture media was switched to serum free media for 24 hr and the media samples were collected for the analysis. Mouse serum was collected from the supernatant of whole blood after centrifugation at 0.4 g for 4 min and analysed using the Mouse High Sensitivity 18-Plex Discovery Assay (Eve Technologies, Calgary, Canada).

## Mass Cytometry

Markers for multiple stress response pathways (*Table 2*) were analysed at single cell level simultaneously by mass cytometry (Helios, Fluidigm). MRC5 cells were treated with either 2 mM or 0.1 mM metformin for 2 days. Positive controls were challenged with either 300 µM $H_2O_2$ in serum-free medium, heat shock (50 °C for 45 min) or starvation (serum-free medium for 24 hr). The cells were trypsinized, washed in PBS, and stained with metal-conjugated antibodies (*Table 2*). Antibodies were either pre-conjugated (Fluidigm), or purified antibodies were conjugated to lanthanide metals using the Maxpar

**Table 2.** List of metal conjugated antibodies for stress response pathway analysis by CyTOF.

| Antibody | Metal | cat # | Vendor |
|---|---|---|---|
| SOD2/MnSOD [9E2BD2] | 176Yb | ab110300 | Abcam |
| GSTM2 (9E975) | 167Er | H00002946-M03 | Novus Biologicals |
| HSF1 | 153Eu | 825,801 | BioLegend |
| Hsp-70 (2A4) | 154Sm | ab5442 | Abcam |
| Nrf2 (phospho S40) [EP1809Y] | 142Nd | ab180844 | Abcam |
| NR1L2/PXR (6H11D8) | 164Dy | LS-C682408-LSP | Stratech Scientific Ltd. |
| GRP78 BiP [EPR4041(2)] | 161Dy | ab108615 | Abcam |
| EIF2S1 (phospho S51) (E90) | 169Tm | ab214434 | Abcam |
| ATF-6 (-Carboxy terminal end) | 175Lu | ab62576 | Abcam |
| GADD153/CHOP | 141Pr | NBP2-13172 | Novus Biologicals |
| CLPP [EPR7133] | 165Ho | ab236064 | Abcam |
| Hsp-60 (LK1) | 144Nd | ab212454 | Abcam |
| pS6 [S235/S236] | 172Yb | 3172008 A | Fluidigm |
| p21 Waf1/Cip1 | 159Tb | 3159026 A | Fluidigm |
| Phospho-Acetyl-CoA Carboxylase (Ser79) (10HCLC) | 170Er | 711,289 | ThermoFisher |
| p62 /SQSTM1 (C-terminus) | 146Nd | GP62-C | Progen |
| p-p38 [T180/Y182] | 156Gd | 3156002 A | Fluidigm |
| DNA/RNA damage | 173Yb | ab62623 | Abcam |
| pHistone H2A.X [Ser139] | 147Sm | NB100-384 | Novus Biologicals |
| PHB | 151Eu | NBP2-32305 | Novus Biologicals |
| p16INK4 | 174Yb | ab54210 | Abcam |
| Cell-ID Intercalator-Ir—500 µM | | 201,192B | Fluidigm |

antibody labelling kit (as per manufacturer's instructions; DVS Sciences) and were stabilised with an antibody stabilisation solution (Candor Bioscience) (*Table 2*). Cells were stained as described (*Cytlak et al., 2020*). Briefly, cells were first stained with 2.5 µM Cisplatin (Fluidigm #201064) for 5 min in PBS for live/dead cell discrimination and washed promptly using Wash buffer (PBS containing 2% FBS). Then the cells were fixed using 1.6% formaldehyde in a working fixation buffer (eBioscience Foxp 3 fixation permeabilisation kit, ThermoFisher Scintific #00–5523) for 30 min, and washed twice with eBioscience perm buffer. Cells were stained in perm buffer for 1 hr with the antibody cocktail containing each intracellular antibody (approximately 0.5 µg in 100 µl per sample) for 1 hr at room temperature and washed twice with PBS. Finally, the cells were fixed with 1.6% formaldehyde in PBS with a nuclear marker, 125 nM iridium (Cell-ID Intercalator-Ir, Fluidigm #201,192B) for 1 hr, and washed using Wash buffer for overnight storage at 4 °C. Prior to CyTOF acquisition, cells were washed twice in 200 µL MilliQ water (600xg for 5 min), counted, diluted to a maximum final concentration of $0.55 \times 10^6$/ml in MilliQ water, and filtered through a 40 µm filter (BD). EQ beads were added (10% by volume) and $0.1 \times 10^6$ cells per sample were acquired on the Helios mass cytometer running CyTOF software v 6.7.1014. The data were analysed using FCS Express 7 (De Novo Software).

## Statistics

Data were analysed with Microsoft Excel and Prism software (GraphPad). Values were expressed either as means with error bars representing SDs or as boxplots with median, upper, and lower quartiles (boxes) and percentiles (whiskers). Graphs were overlaid with the values of all individual biological replicates. Linear regressions and survival curves show means and 95% confidence intervals. Depending on results of normality testing, groups were compared using unpaired t test, Mann-Whitney or one-way

analysis of variance with Tukey post-hoc test. Statistical significance is indicated as *p<0.05, **p<0.01, ***p<0.001, **** p<0.0001.

## Acknowledgements

This study was supported by Cancer Research UK (https://www.cancerresearchuk.org/) Pioneer Award C12161/A24009 and P&G/BBSRC (https://www.bbsrc.ukri.org/) grant BB/S006710/1 to TvZ, an MRC – Arthritis Research UK, Centre for Integrated Research into Musculoskeletal Ageing (https://www. cimauk.org/) Translation grant and UK SPINE Bridge (https://www.kespine.org.uk/) grant B06 to SM and TvZ, a P&G/BBSRC DTP (https://aka.bbsrc.ukri.org/skills/investing-doctoral-training/dtp) studentship (BH174490) to VIK and a BBSRC DTP (https://aka.bbsrc.ukri.org/skills/investing-doctoral-training/dtp) studentship to DJ. The funders had no role in study design, data collection and analysis, decision to publish, or preparation of the manuscript. We thank the Comparative Biology Centre at Newcastle University, in particular Mr. Christopher Huggins, for their expert support with animal husbandry, Drs Andrew Filby and David McDonald at Flow Cytometry Core Facility, Newcastle University for their expert support with CyTOF experiments, and Dr I Karakesisoglou and Dr P Chazot, Durham University, for their support in early supervision of MW.

## Additional information

### Competing interests

Abbas Ishaq: is affiliated with Alcyomics Ltd. The author has no financial interests to declare. The other authors declare that no competing interests exist.

### Funding

| Funder | Grant reference number | Author |
|---|---|---|
| Cancer Research UK | C12161/A24009 | Thomas von Zglinicki |
| Biotechnology and Biological Sciences Research Council | BB/S006710/1 | Thomas von Zglinicki |
| UK SPINE Bridge | B06 | Thomas von Zglinicki Satomi Miwa |
| Biotechnology and Biological Sciences Research Council | BH174490 | Viktor I Korolchuk |
| Biotechnology and Biological Sciences Research Council | | Diana Jurk |

The funders had no role in study design, data collection and interpretation, or the decision to submit the work for publication.

### Author contributions

Edward Fielder, Data curation, Formal analysis, Investigation, Validation, Visualization, Writing – original draft, Writing – review and editing; Tengfei Wan, Data curation, Formal analysis, Investigation, Visualization; Ghazaleh Alimohammadiha, Evon Low, B Melanie Weigand, George Kelly, Craig Parker, Brigid Griffin, Investigation; Abbas Ishaq, Formal analysis, Investigation; Diana Jurk, Funding acquisition, Supervision; Viktor I Korolchuk, Funding acquisition, Investigation, Supervision; Thomas von Zglinicki, Conceptualization, Data curation, Formal analysis, Funding acquisition, Investigation, Methodology, Project administration, Supervision, Writing – original draft, Writing – review and editing; Satomi Miwa, Conceptualization, Formal analysis, Funding acquisition, Investigation, Project administration, Writing – original draft, Writing – review and editing

### Author ORCIDs

Edward Fielder http://orcid.org/0000-0003-2834-8706

Thomas von Zglinicki http://orcid.org/0000-0002-5939-0248

### Ethics

All animal experimentation was performed in compliance with the guiding principles for the care and use of laboratory animals (ARRIVE guidelines). The study was licenced by the UK Home Office (PB048F3A0).

### Decision letter and Author response

Decision letter https://doi.org/10.7554/eLife.75492.sa1
Author response https://doi.org/10.7554/eLife.75492.sa2

---

## Additional files

### Supplementary files

• Transparent reporting form

### Data availability

All data generated or analysed during this study are included in the manuscript and supporting files; Source Data files have been provided for all Figures.

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
