## [Editor Report]

This is an exciting study with translational implications that patients exposed to radiation as a form of therapy for cancer may avoid complications down the road by utilising medication to eliminate the senescent cells generated by the treatment and improve quality of life.

---

## [Decision Letter]

**Decision letter after peer review:**

Thank you for submitting your article "Short senolytic or senostatic interventions rescue progression of radiation-induced frailty and premature ageing in mice" for consideration by *eLife*. Your article has been reviewed by 2 peer reviewers, and the evaluation has been overseen by a Reviewing Editor and Carlos Isales as the Senior Editor. The reviewers have opted to remain anonymous.

The Reviewing Editor has drafted this to help you prepare a revised submission.

Essential revisions:

1) Please provide the rationale for the approaches used in the introduction. This includes the specific senolytic therapies employed. Potentially consider reducing the emphasis on metformin to introduce the reasons for selecting specific tissues, drug doses etc. more fully.

2) Add consideration of the ages of mice used in the limitations section with some discussion of potential benefits of studies in older and younger animals to better model conditions in humans.

3) While it is understandable that the authors have not yet completed studies on females, this should be left as a limitation. The reference to the authors' "in preparation" companion work on females, while important in the longer term, adds little to this study. It would be worth expanding the discussion of sexual dimorphism in lifespan interventions to include healthspan extending interventions.

4) Some figures/data points are difficult to see. For example, the data points on the frailty index versus age plots are hard to see. Please review all the figures to ensure the data points are legible.

5) Similarly, some of the fonts used on a number of the figure panels are very small and hard to see. This should be revised so that the text in all figure panels is legible.

6) I suggest the authors include more data to evaluate the pathways targeted by the senolytic and senostatic agents, mechanistically evaluate how the senolytic and senostatic treatments are affecting the SASPs. Changes in functional indicators measured in the in-vivo experiments could be an outcome of several pathways, to test the hypothesis that this was indeed mediated through the reduction of SASPs , those specific markers should be evaluated in all the experiments. It is also critical to include a head to head comparisons of the senolytic and senostatic treatment.

---

## [Author Response]

Essential revisions:1) Please provide the rationale for the approaches used in the introduction. This includes the specific senolytic therapies employed. Potentially consider reducing the emphasis on metformin to introduce the reasons for selecting specific tissues, drug doses etc. more fully.

Rationales for the specific senolytics and for the focus of our assays with respect to functional domains and tissues are now given in the introduction. Drug doses have been justified in the Results sections 1 (Navitoclax and D+Q) and 3 (Metformin). Please see also answers to reviewer 1 points 1-4 below.

2) Add consideration of the ages of mice used in the limitations section with some discussion of potential benefits of studies in older and younger animals to better model conditions in humans.

Has been added to the limitations section.

3) While it is understandable that the authors have not yet completed studies on females, this should be left as a limitation. The reference to the authors' "in preparation" companion work on females, while important in the longer term, adds little to this study. It would be worth expanding the discussion of sexual dimorphism in lifespan interventions to include healthspan extending interventions.

We have expanded the discussion citing novel reviews on sexual dimorphism in ageing and extended it to include healthspan-targeting interventions. The reference to the companion work has been deleted.

4) Some figures/data points are difficult to see. For example, the data points on the frailty index versus age plots are hard to see. Please review all the figures to ensure the data points are legible.

All plots have been redrawn using increased fonts were necessary. To improve clarity of the frailty plots, figures with increased symbol size were added as supplements to Figures 1B, 2B and 3B.

5) Similarly, some of the fonts used on a number of the figure panels are very small and hard to see. This should be revised so that the text in all figure panels is legible.

Figures 1 to 6 and the supplementary figures were revised and larger fonts used were necessary.

6) I suggest the authors include more data to evaluate the pathways targeted by the senolytic and senostatic agents, mechanistically evaluate how the senolytic and senostatic treatments are affecting the SASPs. Changes in functional indicators measured in the in-vivo experiments could be an outcome of several pathways, to test the hypothesis that this was indeed mediated through the reduction of SASPs , those specific markers should be evaluated in all the experiments. It is also critical to include a head to head comparisons of the senolytic and senostatic treatment.

We do not claim in the paper that changes in functional indicators measured in the in-vivo experiments were mediated through the reduction of SASPs. What we claim and show is that both senolytic and senostatic interventions reduce senescent cell frequencies together with multiple functional outcomes over the lifecourse. We agree that the impact of senescent cell reduction onto these functional improvements could be mediated by different pathways that might be more or less tightly related to the SASP. These pathways are probably tissue- and cell-type specific. Assessing all these would in our opinion go far beyond what can be expected from a single paper.

When assessing the senostatic activity of metformin in vivo, we claim and show that it reduces senescent ROS and the SASP, and we show the pathway that leads to it. We and others have shown previously that reducing ROS and SASP production from senescent cells reduces bystander senescence. Together, this identifies a pathway by which metformin at physiologically achievable concentrations reduces senescent cell frequencies.

In order to link our data more closely to SASP, we have measured levels of 18 cytokines/chemokines that are part of the SASP at the end of the experiment in the serum of senolytic- and metformin-treated mice. These data are now integrated into results part 1 and 3. They show that at one year after senolytic intervention, there is no remaining difference in the measured SASP component levels. However, the longer-lasting metformin intervention still results in a persistent tendency for reduction of some SASP components, notably including IL17 and TNFa (albeit at only p=10%), which were also found reduced in vitro (Figure 4E), together with the prominent SASP component CCL2 (at p<0.05).

We did discuss carefully the question of presenting our senolytics vs metformin data in a head-to-head format, e.g. combining the data in Figures1 and 3 in the same graphs. The outcome is that we do not believe that this is the appropriate presentation for our results. We do show already the Navitoclax vs D+Q data head-to-head, because they were generated using a single common sham control group. However, metformin was given to the animals by a different route (in soaked food instead of gavage) in accordance with widespread practice. This required a separate control group also receiving soaked food, which resulted in higher food intake, greater body weight and somewhat different capabilities in the neuromuscular tests in the metformin control as compared to the senolytics control (most probably due to differences in body weight between the control groups). Therefore, a head-to-head comparison of all groups would distract from the essential information, e.g. the intervention effects. We have tried to make the comparison between the senolytic and senostatic interventions as easy as possible by presenting data in Figures1 and 3 and their associated supplements as similarly as possible, but do think that a direct head-to-head comparison would not be correct for these two independently designed experiments.